# Polymeric Nanoparticles as Tunable Nanocarriers for Targeted Delivery of Drugs to Skin Tissues for Treatment of Topical Skin Diseases

**DOI:** 10.3390/pharmaceutics15020657

**Published:** 2023-02-15

**Authors:** Eiman Abdalla Madawi, Alaa Raad Al Jayoush, Mutasem Rawas-Qalaji, Hnin Ei Thu, Shahzeb Khan, Mohammad Sohail, Asif Mahmood, Zahid Hussain

**Affiliations:** 1Department of Pharmaceutics and Pharmaceutical Technology, College of Pharmacy, University of Sharjah, Sharjah 27272, United Arab Emirates; 2Research Institute for Medical and Health Sciences, University of Sharjah, Sharjah 27272, United Arab Emirates; 3Dr. Kiran C. Patel College of Allopathic Medicine, Nova Southeastern University, Fort Lauderdale, FL 33326, USA; 4Research and Innovation Department, Lincoln University College, Petaling Jaya 47301, Malaysia; 5Department of Pharmacy, University of Malakand, Chakdara 18800, Pakistan; 6Center for Pharmaceutical Engineering Science, Faculty of Life Sciences, School of Pharmacy and Medical Sciences, University of Bradford, Bradford BD7 1DP, UK; 7Discipline of Pharmaceutical Sciences, School of Health Sciences, University of KwaZulu-Natal, Private Bag X54001, Durban 4000, South Africa; 8Faculty of Pharmacy, Cyprus International University, Nicosia 99258, Cyprus; 9Department of Pharmacy, University of Chakwal, Chakwal 48800, Pakistan

**Keywords:** polymeric nanoparticles, topical delivery, skin diseases, psoriasis, wound healing, skin cancer, atopic dermatitis

## Abstract

The topical route is the most appropriate route for the targeted delivery of drugs to skin tissues for the treatment of local skin diseases; however, the stratum corneum (SC), the foremost layer of the skin, acts as a major barrier. Numerous passive and active drug delivery techniques have been exploited to overcome this barrier; however, these modalities are associated with several detrimental effects which restrict their clinical applicability. Alternatively, nanotechnology-aided interventions have been extensively investigated for the topical administration of a wide range of therapeutics. In this review, we have mainly focused on the biopharmaceutical significance of polymeric nanoparticles (PNPs) (made from natural polymers) for the treatment of various topical skin diseases such as psoriasis, atopic dermatitis (AD), skin infection, skin cancer, acute-to-chronic wounds, and acne. The encapsulation of drug(s) into the inner core or adsorption onto the shell of PNPs has shown a marked improvement in their physicochemical properties, avoiding premature degradation and controlling the release kinetics, permeation through the SC, and retention in the skin layers. Furthermore, functionalization techniques such as PEGylation, conjugation with targeting ligand, and pH/thermo-responsiveness have shown further success in optimizing the therapeutic efficacy of PNPs for the treatment of skin diseases. Despite enormous progress in the development of PNPs, their clinical translation is still lacking, which could be a potential future perspective for researchers working in this field.

## 1. Introduction

Skin diseases affect millions of people worldwide, representing a significant part of the global total of diseases [1]. They have several etiologies including inflammation, infection, or tumors, and can involve some or all of the skin layers. The non-topical administration of drugs has been conventionally used for the treatment of skin diseases; however, there are limitations that restrict its use, including the insufficient amount of drugs reaching skin tissues and their accumulation elsewhere resulting in the development of off-target adverse events [2]. To prevent this from happening, the topical route has been extensively investigated as a most promising port of drug delivery; however, there are substantial barriers to the topical administration of drugs that need to be addressed and overcome to improve the therapeutic outcomes of drugs administered through the topical route.

### 1.1. Skin Structure

Compared to other body organs, the skin is considered as the largest organ covering the largest surface area of the body, which is approximately 1.8–2.0 m^2^ [3,4]. Being the foremost protective barrier to the environment, the skin plays a significant role in defending the body against various pathogens, chemicals, and radiation [4,5]. The skin is made up of three main layers: (1) the epidermis, which is the upper or external layer; (2) the hypodermis, which is the deepest layer; and (3) the dermis, which resides in between the epidermis and the hypodermis [3,4,5,6] (Figure 1A). The epidermis further comprises five distinct layers, among which the outermost layer is known as the stratum corneum (SC) followed by the stratum lucidum, stratum granulosum, stratum spinosum, and stratum basale (Figure 1B). The SC serves as a major outermost barrier to all permeants (e.g., drugs, chemicals, radiation, pathogens, etc.) which tend to permeate through the skin. The intracellular matrix of SC is rich in lipids such as ceramides, free fatty acids, and cholesterol, which provides a hydrophobic lamellar structure that prominently contributes to the barrier properties of SC [7,8]. Furthermore, four major types of cells constitute the epidermis: (1) keratinocytes (dead), which are responsible for forming a water-proofing barrier and regulate body temperature and prevent transepidermal water loss; (2) melanocytes, which have a major function in producing melanin, which acts as a protective sunscreen barrier against UV radiation [4,5]; (3) Langerhans cells, which are one of the immune cells that assist in presenting and eradicating antigens (e.g., pathogens); and (4) Merkel cells, which are responsible for the sensory functions of the skin [5]. Beneath the epidermis, a thick dermal layer exists which further comprises two different layers of connective tissue depending upon the thickness of collagen in each layer. The upper dermal layer is thin and is called the papillary dermis, and the lower dermal layer is known as the reticular dermis, which is relatively thicker due to a higher thickness of collagen compared to the papillary dermis. In addition, the dermis contains many complimentary structures such as oil and sweat glands, hair follicles, nerves, blood vessels, and other structures [3,4,5,6]. The third layer is the hypodermis, the deepest layer of the skin and commonly known as subcutaneous tissue, which is primarily made up of rich fatty contents (adipose tissue) and some connective tissue. Some appendages are found in the hypodermis such as blood vessels, nerves, and hair follicles. The thick fatty layer of the hypodermis plays numerous roles such as serving as an energy reservoir because of its high fat contents, while the fat lobules in this layer play a significant role in isolating the body and tolerating external mechanical pressure by acting as cushions [3,4,5,6].

### 1.2. Topical Drug Delivery through the Skin

Drug molecules need to pass through the skin, as the main anatomical barrier, to exert their pharmacological effects; therefore, many studies have been conducted to determine how topically applied drugs can penetrate through the skin [9]. Applied drugs have two main pathways for their percutaneous penetration. First is the intercellular or intracellular pathway to cross the SC followed by the diffusion or partitioning of drugs through the viable parts of the epidermis and the dermis. Second is the appendageal pathway [10] that allows for the passage of drugs through sweat glands and hair follicles [11]; however, the cumulative surface area of this route represent approximately 0.1% of the entire skin [9]. Depending upon the physicochemical properties of the applied drug molecules, they can permeate through either or both pathways on the skin [10].

Drug delivery through the skin has a great potential since the skin has the largest surface area in the human body. It can be employed for topical, dermal, or transdermal drug delivery. Topical delivery retains pharmaceutical ingredients on the upper layers of the skin, while in dermal delivery, the drug penetrates to the dermal layer, and in transdermal delivery, it becomes absorbed deeply into the skin layers, reaching systemic circulation [12]. Topical and dermal delivery are mainly used to induce local effects in the skin that bring along many advantages such as minimizing the dose required to reach to the site of action (skin layers) compared with oral or parenteral administration, thus minimizing the dose-dependent systemic side effects and toxicity. On the other hand, transdermal delivery aims to deliver applied drugs through the skin for systemic targeting [13]. For both of these delivery routes, SC is considered as the rate-limiting barrier to drug penetration through the skin [10]. Generally, drugs permeation through the SC follow Fick’s second law [13]:(1)$$J=\frac{DCP}{L}$$
where *J* = drug flux, *D* = drug diffusion coefficient, *C* = drug concentration, *P* = drug partition coefficient, and *L* = SC thickness.

Generally, small molecules (<500 Daltons) with moderate lipophilicity (partition coefficient of 10–1000) can easily penetrate through the SC and reach the deeper skin layers [10]; however, relatively larger molecules such as proteins and DNA are difficult to be delivered through the skin and thus effective solutions and strategies are needed to overcome their limited permeability [13]. Several strategies have been adapted to facilitate the penetration of macromolecular drugs through the skin; they are divided into active and passive methods [10].

Active enhancement methods include the use of electrical force (e.g., iontophoresis and electrophoresis), mechanical forces (e.g., microneedle array), or other forces (e.g., ultrasound, magnetic field, temperature, etc.). On the other hand, passive methods include temporary disruption of the integrity of SC via hydration or chemical enhancers, drug optimization (chemical modification of drug structure such as prodrug formation), and the use of specialized delivery systems such as polymeric nanoparticles (PNPs), solid lipid nanoparticles, nanoemulsion, and nano-vesicular carriers [10].

All these techniques have shown promising results in improving the delivery of drugs through the SC; however, this review will focus on critically discussing the pathogenesis of various skin diseases, conventional treatments and their limitations, and the therapeutic superiority of natural PNPs for the treatment of various dermatological diseases such as psoriasis, atopic dermatitis, skin cancer, acute and chronic wounds, skin infections, and acne. To the best of our knowledge, this is the first article discussing the use of PNPs with special emphasis on PNPs prepared from natural polymers for the topical management of different skin diseases and conditions.

## 2. Physicochemical Properties and Types of PNPs

Nanotechnology has revolutionized many fields, including the medical and health sciences. In the medical field, nanotechnology has attracted considerable interests in drug delivery, diagnosis, and theranostic applications due to the inherent ability of nano-delivery systems to (I) protect the drug from the environment by physical and chemical means, (II) control and sustain the release of cargo, (III) improve the solubility of hydrophobic drugs, (IV) enable a targeted delivery to specific cells and tissues by functionalizing their surfaces with cell-specific ligands, and (V) preferentially accumulate into target tissues due to the enhanced permeability and retention (EPR) effect [10,13].

Among the different types of nanoparticles being employed in the medical field, PNPs have been extensively investigated for their ability to improve drug delivery through the skin, where they tend to localize in the hair follicles and release their cargo, resulting in significantly enhanced drug permeation through different skin layers [13]. PNPs can be synthesized from a variety of polymers that are either natural, such as chitosan, alginate, and guar gum, or synthetic, such as poly (lactic-co-glycolic acid) (PLGA) and poly-ε-caprolactone (PCL). In this review, we have mainly focused on the biomedical significance of PNPs synthesized from natural polymers to treat different types of skin diseases.

PNPs exhibits nano-scaled dimensions (1–1000 nm) and may contain drug(s) either encapsulated within the polymeric core or uniformly dispersed within the polymeric matrix [14]. Depending on their morphology and architecture, PNPs can be classified as nanocapsules or nanospheres (Figure 2). Nanocapsules are composed of a polymeric shell, which is absent in nanospheres, entrapping either an oily or aqueous liquid dissolving the drug [15,16]. Accordingly, they act as a reservoir system, where the drug first partitions into the membrane then diffuses through the membrane to the surrounding medium. The concentration gradient that is kept between the reservoir and the medium is responsible for fluxing the drug outside the system at a constant rate, thus achieving a zero-order kinetics [15]. On the other hand, in nanospheres, the polymer forms a network in which the drug is dissolved, encapsulated, or adsorbed, and this is recognized as a matrix system (known also as a monolithic system). Depending upon the nature and characteristics of the polymer used for the preparation of PNPs, a drug is released through a diffusion process either due to the desorption of the adsorbed drug, swelling of the polymeric matrix in response to the pH, temperature, light, magnetic field, etc., or the degradation of the polymeric matrix [15,17]. Generally, drugs adsorbed on the surface of PNPs results in an initial burst release followed by a relatively slow and sustained release of the drug from the polymeric matrix or the core [14].

## 3. Topical Application of PNPs for Treatment of Skin Diseases

### 3.1. Natural Polymers Used to Prepare PNPs for Treatment of Skin Diseases

Various natural polymers have been harnessed to prepare PNPs for the management of dermatological diseases. Chitosan (CS) is a biocompatible and biodegradable natural polysaccharide that has been extensively utilized to synthesize PNPs for drug delivery purposes [18]. Under acidic conditions, the amino groups of CS undergo protonation, resulting in positive charge formation, thus making this polymer cationic for a better electrostatic interaction with negatively charged biological surfaces such as epithelial membranes, which ultimately results in fluidization and the opening of tight junctions in the epithelium. This mucoadhesive property of CS improves the topical delivery of drugs via prolonging the drug residential time at the site of application as well as promoting a better permeation through the SC due to opening tight junctions within the epithelium [19]. Fereig et al. [20] prepared tacrolimus (TAC)-loaded CS-NPs by the ionic gelation method. TAC was solubilized in an organic co-solvent and pentasodium tripolyphosphate anhydrous (TPP) was added to the co-solvent solution. After that, the prepared mixture was injected into the CS solution slowly, which was followed by stirring [20].

Another naturally originated polymer, alginate (ALG), has also been used for the preparation of PNPs for the treatment of skin conditions. ALG is a naturally occurring polysaccharide that is derived from algae [21]. Due to the carboxylic functional groups present in this polymer, ALG is of an anionic nature, providing it with mucoadhesive properties that surpass other cationic and neutral polymers [22]. Ionic interactions between ALG and polycationic polymers such as CS may result in more stable nanomaterials; thus, CS/ALG-NPs have been extensively exploited as versatile drug delivery systems for the treatment of various skin diseases [23,24]. Gomez et al. [25] prepared curcumin (CUR)-loaded CS/ALG-NPs by emulsification, followed by the ionotropic gelation method. O/W emulsion was first prepared by the addition of organic CUR solution to ALG and surfactant aqueous solution. After sonication, the CS solution was added to the polyanionic emulsion forming the NPs suspension [25].

Hyaluronic acid (HA) is another natural polymer that is used in skin drug delivery. It has been found to enhance drugs’ skin permeation by penetrating the SC via transcellular transport [26]. It has been employed in skin diseases by the fabrication of PNPs [27] or in the functionalization of other types of PNPs [28]. Tolentino et al. [27] used the ionic gelation method to prepare HA NPs loaded with clindamycin. A solution of poly-L-arginine hydrochloride was dripped into a HA and clindamycin dispersion in purified water followed by stirring and sonication [27].

Other examples of natural polymers that have been utilized to prepare PNPs include guar gum (GG) and xanthin gum (XG) either alone or in combination with other types of polymers [29,30]. Ghosh et al. [31] synthesized GG NPs by an acid hydrolysis reaction. Aqueous sulfuric acid was added to GG while stirring at a high temperature. The prepared mixture was then poured into water to cool it down and was followed by dialysis for 72 h [31]. A summary of other natural polymers used to prepare PNPs for the treatment of dermatological diseases is presented in Table 1.

### 3.2. Psoriasis

#### 3.2.1. Pathogenesis of Psoriasis

Psoriasis is a chronic autoimmune skin inflammatory disease affecting 0.9–11.4% of the world’s population according to the World Health Organization (WHO) [57]. It is manifested by itchy and scaly red lesions with silvery raises present on the skin, scalp, knees or elbows, or yellow-white patches on the tongue, and they may spread according to the disease’s severity [58]. The exact causes are not fully known but they can be linked to stress, skin injuries caused by trauma, sun exposure, alcohol consumption, cigarette smoking, and/or family history [59]. One of the pathways involves the secretion of antimicrobial peptides (AMPs) from the keratinocytes that become overexpressed in response to trauma. AMPs comprise LL37, S100 proteins, and β-defensins. As a result, pro-inflammatory mediators such as interleukins (ILs), interferons (IFNs), and tumor necrosis factor-α (TNF-α) are released and activate TH-1 and TH-7 T-lymphocytes, which ultimately triggers an adaptive immune response and keratinocytes proliferation in the epidermis [60,61].

#### 3.2.2. Conventional Treatments for Psoriasis and Their Limitations

Conventional management of psoriasis combines four main approaches: (I) systemic therapy, (II) topical therapy, (III) phototherapy, and (IV) biological systemic therapy [59]. Drugs taken by the systemic routes are mostly required in case of moderate-to-severe conditions. They include methotrexate (MTX), cyclosporine (CYP), acitretin, and fumaric acid esters. MTX is a folic acid antagonist that inhibits DNA synthesis [61]. The most common side effects include nausea, vomiting, loss of appetite, and hepatotoxicity [62]. CYP is an effective immunosuppressant that acts as a remission inducer and for maintenance therapy in psoriasis [61]; however, due to its cardiovascular, renal, endocrinological, and neurological toxicity, it has been limited to short-term therapy only [63]. Acitretin is a retinoid that is also used in the treatment of psoriasis in very low doses due to its side effects, such as dry skin, conjunctivitis, hair loss, and teratogenicity [64]. Fumaric acid esters are used in psoriasis management as well due to their immunomodulatory, anti-inflammatory, and anti-oxidative properties [65,66]. Their common side effects include gastrointestinal manifestations in addition to lymphocyte and leukocyte counts reduction [67].

In mild-to-moderate conditions, topical treatment is the first line therapy, which is also used in combination with the other three approaches in case of moderate-to-severe conditions [59]. Topical delivery is preferred over systemic administration due to its lower systemic toxicity and immunosuppression. Most used drugs for the treatment of psoriasis include corticosteroids, vitamin A and D analogues, dithranol, and immunosuppressants such as TAC; however, poor drug penetration through the skin remains a main obstacle to conventional topical therapies [68].

Alternatively, phototherapy is also an appropriate option that has shown a good efficacy against moderate-to-severe psoriasis. Different light sources have been studied such as ultraviolet B (UVB), psoralen ultraviolet A (PUVA), pulsed dye laser (PDL), photodynamic therapy (PDT), intense pulsed light (IPL), and light-emitting diodes (LED) [69]. Among these, UVB and PUVA with a wavelength of 290–320 nm and 320–400 nm, respectively, are preferred due to their low cost and relatively greater safety and efficacy [69]. The mechanism through which phototherapy relieves psoriatic symptoms is believed to be by the modulation of the cytokine profile, the induction of apoptosis in hyper-proliferative keratinocytes, and enhanced immunosuppression [70]. Notably, despite its promising anti-psoriatic potential, phototherapy is also associated with numerous side effects such as erythema, photo-aging, and melanoma, which limits their clinical acceptability [69].

Biologics is another promising therapy for the treatment of psoriasis. These are macromolecular proteins which are intended to be administered systematically to achieve a better therapeutic efficacy due to their susceptibility to degradation by gastric pH and enzymes [71]. The most suggested mechanisms by which biologics treat psoriasis include an inflammatory pathway involving TNF-α (e.g., etanercept, infliximab, adalimumab, and certolizumab) and IL-23/Th17 axis (e.g., ustekinumab, guselkumab, secukinumab, and brodalumab). Nevertheless, biologics exhibit a reasonable efficacy against psoriasis; however, due to their strong immunosuppressive potential, they are used as a last line of therapy [61].

Despite the fact that conventional treatments have shown a satisfactory outcome in the treatment of psoriasis, due to the lack of balance between their efficacy and safety, researchers are constantly looking for alternate therapeutic regimens with a superior therapeutic efficacy with minimal side effects. Thus, PNPs have been extensively exploited due to their inherent and distinctive features, as one of the most promising carriers for topical drug delivery for the treatment of psoriasis. PNPs do not only facilitate the superior permeation of drugs through the SC but also exhibit prolonged retention into various skin layers with minimal systemic toxicity (Figure 3).

#### 3.2.3. PNPs-Based Topical Therapies for Treatment of Psoriasis

A wide variety of PNPs have been synthesized from a wide range of natural polymers and have been tested for the treatment of psoriasis or psoriatic lesions. Among the various natural polymers, CS is the most exploited polymer for the fabrication of PNPs used against psoriasis. Terzopoulou et al. [32] prepared CS-NPs loaded with CUR, a naturally occurring polyphenol with potent anti-inflammatory activity. CUR-encapsulated CS-NPs were further incorporated into a collagen-based patch and this hybrid delivery vehicle was then tested for the treatment of psoriasis. The great swellability and mucoadhesive potential of a fabricated patch while maintaining a good hydration level at psoriatic lesions eventually provide a symptomatic relief for itchiness in psoriatic patients. Furthermore, in vitro testing on human keratinocytes obtained from psoriatic patients displayed a significant repression in proliferation that eventually results in a reduction in hyperkeratosis [32]. This was attributed to the synergistic anti-proliferative and anti-mitotic activity of CUR and CS as well as an excellent permeability efficiency of PNPs that inhibited mitosis in human keratinocytes [72].

Similarly, Chamcheu et al. [33] studied the effectiveness of green tea polyphenol (EGCG)-loaded CS-NPs against imiquimod (IMQ)-induced skin inflammation in BALB/c mice (Figure 4). A visual examination of mice ears showed a superior effectiveness of topically administered EGCG-loaded CS-NPs compared to free EGCG and the IMQ-induced positive control group with no treatment (Figure 4A–D). The treatment efficacy was assessed in terms of reduced erythema (Figure 4M), scaling (Figure 4N), and ear and skin thickness (Figure 4O,P) which were more prominent in case of psoriatic lesions treated with PNPs. These results were also evident from the histological examination of treated psoriatic lesions in terms of acanthosis, epidermal rete-ridge projections, and Munro’s micro abscesses (Figure 4E–L). In addition, cells hyper-proliferation was assessed by analyzing the Ki67+ cells in the full-thickness skin and the epidermis, revealing a lower cells expression in EGCG-CS-NPs compared to free EGCG and the IMQ-induced positive control group (Figure 4Q) [33]. These results clearly demonstrate the superior efficacy of PNPs for the treatment of psoriatic lesions in experimental animals.

In another study, Fereig et al. [20] encapsulated tacrolimus (TAC) in CS-NPs and studied their deposition efficiency in skin tissues. Comparative analysis showed a significantly higher TAC retention into various skin layers in case of TAC-CS-NPs compared to marketed TAC ointment. Moreover, the superiority of PNPs was also assessed in terms of psoriatic skin recovery. Interestingly, the results showed that psoriatic lesions treated with TAC-CS-NPs displayed enhanced hair growth within only three days of treatment compared to the control group [20]. In a similar fashion, Shandil et al. [34] fabricated CS-NPs containing gallic acid and rutin then coated them with Tween-80. In vitro studies on HaCaT cells (human keratinocytes) showed a significant repression in their proliferation and good antioxidant, anti-inflammatory, and antimicrobial efficacy, which suggested their potential usage in psoriasis management [34]. In another study published by Cordenonsi et al. [35], CS was used as a coating material to functionalize the surface of nanostructured lipid carriers (NLCs) loaded with fucoxanthin (FUCO). The results suggested a significantly higher cell uptake efficiency and a reduction in the cell viability and proliferation in normal dermal human fibroblasts treated with CS-coated FUCO-loaded NLCs [35].

ALG is another natural polymer that has been used for the preparation of PNPs for the treatment of psoriasis. Gomez et al. [25] studied the anti-psoriatic efficacy of combined therapy involving CUR-loaded PNPs (CS/ALG-NPs) and phototherapy against TNF-α-induced keratinocytes. A synergistic anti-proliferative activity was evidenced in keratinocytes treated with combined therapy involving CUR-CS/ALG-NPs with phototherapy in contrast to free CUR. The synergistic efficacy of combined therapy was attributed to the enhancement of the cellular uptake of CUR-CS/ALG-NPs, sustaining the release of CUR, the protection of encapsulated CUR from premature degradation, and the anti-proliferative effect of phototherapy. These findings signify the combined treatment regimen containing CUR-loaded CS/ALG-NPs and phototherapy for the treatment of psoriasis [25].

### 3.3. Atopic Dermatitis

#### 3.3.1. Pathogenesis of Atopic Dermatitis

Atopic dermatitis (AD) is a common chronic skin inflammatory disease characterized by erythematous and itchy skin with an undergoing inflammatory response that can affect individuals in their early, mid, or late stages of life. Patients are manifested by red, dry skin lesions, with papules, exudate, and severe pruritis [73]. The exact pathogenesis of AD is still unknown, but it is believed that AD can be caused by a combination of factors such as genetic mutation, environmental factors, and a defective skin barrier [74]. Primarily, mechanical injury that disrupts the natural barrier offered by the SC can trigger the immune response by the activation of mast cells and T-lymphocytes, resulting in increased immunoglobulin-E (IgE) levels and inflammatory signals [75,76,77].

#### 3.3.2. Conventional Treatments for Atopic Dermatitis and Their Limitations

To date, there is no absolute therapy that can completely cure AD; however, certain pharmacological agents are commonly applied to attenuate the severity of symptoms [78]. Topical corticosteroids are considered as the first line of treatment for moderate-to-severe AD. They can be classified according to their potency into (I) mild (e.g., hydrocortisone), (II) moderate (e.g., hydrocortison-17-butyrate and clobetason-17-butyrate), (III) strong (e.g., betamethasone, mometasone furoate, and desoximetasone), and (IV) very strong (e.g., clobetasol propionate). In most of the AD cases, symptoms can be managed with mild and moderate potency corticosteroids [79]; however, they may cause skin thinning and stretch marks, particularly upon their chronic use, which reduces patient’s compliance to these medications [80]. In addition, topical calcineurin inhibitors (TCIs), which are non-steroidal anti-inflammatory agents, can be alternatively used to reduce the severity of skin inflammation and flares, particularly in young children and infants. The most commonly used TCIs include pimecrolimus that is comparable to mild corticosteroids in terms of potency while tacrolimus is comparable to moderate-to-strong corticosteroids [81]. TCIs are also not free of side effects and may cause irritation, erythema, burning, and stinging at the site of application [82].

Similarly to psoriasis, severe AD cases are managed with systemic immunosuppressants such as oral corticosteroids, azathioprine (AZP), CYP, MTX, and mycophenolate mofetil (MMF). However, due to renal, hepatic, and other toxicities, the use of all these agents is limited [83]. In addition, antimicrobials can be used to eradicate infections associated with inflamed skin lesions specifically caused by *S. aureus* [84]. On the other hand, antihistamines are also used to reduce pruritis and sever itching associated with various skin conditions. Phototherapy involving the UVA and UVB is another viable therapeutic option; however, it is associated with many side effects, as discussed earlier in the psoriasis management [85]. Due to these limitations associated with conventional therapeutic agents, efforts have been made to offer alternate viable options to treat mild-to-severe AD with minimal side effects and better patient compliance.

#### 3.3.3. PNPs-Based Topical Therapies for Treatment of Atopic Dermatitis

Many researchers have investigated the therapeutic efficacy of PNPs for the treatment of mild-to-severe AD. Shadab et al. [36] fabricated CS-NPs for the topical delivery of betamethasone valerate (BMV), which is a moderate potency topical corticosteroid. BMV-loaded CS-NPs displayed Fickian diffusion at pH 5.5 (simulated to AD lesion). Moreover, the BMV permeation efficiency as well as the amount retained in the epidermis and the dermis were significantly higher in case of BMV-CS-NPs compared to free BMV in solution form. These findings evidenced the potential of PNPs for the topical delivery of corticosteroids [36]. The permeability and skin retention efficiencies of BMV-loaded PNPs were further improved by functionalizing their surface with HA [37]. Skin permeation and retention studies revealed a significantly higher amount of BMV permeated and retained into the epidermis and dermis from HA-BMV-CS-NPs compared to bare BMV-CS-NPs. These results signify the importance of the functionalization of PNPs with HA for the topical delivery of corticosteroids [37].

Likewise, Siddique et al. [38] designed PNPs (CS-NPs) for the combined topical delivery of hydrocortisone (HC) (mild potency corticosteroid) and hydroxytyrosol (HT) (potent natural polyphenol with a strong antioxidant property) for the treatment of AD. The formulated PNPs (HC-HT-CS-NPs) were tested on human volunteers with healthy skin to measure their safety profile in terms of local and systematic toxicity. No signs of local irritation and redness as well as systemic toxicity were observed in any volunteer, indicating that HC-HT-CS-NPs were safe to be used in AD patients [38]. 

Prior to human testing, these HC-HT-CS-NPs were assessed for their safety and efficacy on AD-induced NC/Nga mice [39]. No signs of toxicity along with a superior control over AD symptoms (erythema intensity, AD scores, and transepidermal water loss) were observed in mice treated with PNPs compared to those treated with free drug or commercial topical medication for AD. The blood chemistry of mice treated with HC-HT-loaded PNPs revealed a significant repression in the release of pro-inflammatory mediators such as IgE, histamine, prostaglandin E_2_ (PGE_2_) and vascular endothelial growth factor-α (VEGF-α) compared to other tested groups. Moreover, the histological examination revealed a faster recovery of skin integrity with regrown elastic connective tissues and a pronounced reduction in fibroblast infiltration, which are key players of skin inflammation [39]. 

The ability of PNPs for a topical combo delivery of HC and HT in 2,4-dinitrofluorobenzene (DNFB)-induced AD NC/Nga was also validated in a mouse model [41]. Lowest skin thickness and ADI (AD index) represented by alleviated erythema, dryness, edema, and erosion were observed in mice treated with HC-HT-CS-NPs compared to commercial AD cream (DermAid^®^) and other tested formulations (Figure 5) [41]. The blood chemistry of mice treated with HC-HT-CS-NPs showed a significant downregulation in the expression of IgE, histamine, PGE2, VEGF-α, and T helper cell type 1 and 2 (Th1, Th2) cytokines. These findings were also supported by a histological examination that showed lower keratinized epithelium fragmentation, acanthosis, the number of inflammatory cells infiltrated in the dermis, and hyperkeratosis [41].

In another study, Zhuo et al. [42] fabricated TAC-loaded CS-NPS coated with HA and investigated in an AD mouse model. Interestingly, the data obtained from ex vivo permeation studies using full-thickness NC/Nga mouse skin demonstrated a lower penetration of TAC from HA-TAC-CS-NPs compared to TAC-CS-NPs. The reported slower permeation of TAC through the full-thickness mouse skin in case of HA-TAC-CS-NPs was explained due to the strong mucoadhesion of HA-functionalized PNPs on skin epithelium, resulting in the sustained release of TAC. This achieved a sustained release and permeation could be beneficial for prolonging efficacy while reducing the frequency of required topical applications. In addition, it was demonstrated that HA-functionalized PNPs exhibited a significantly higher deposition into epidermis and dermis compared to bare PNPs and commercial formulations. Finally, testing HA-TAC-CS-NPs incorporated into a cream formulation against AD-induced NC/Nga mice displayed a superior anti-AD efficacy that was assessed in terms of lower transepidermal water loss (TEWL), erythema intensity, and ADI scores [42].

Another natural polysaccharide, GG, which is derived from the seeds of *Cyamopsis tetragonoloba* plant, has also been tested for the treatment of AD [86]. Ghosh et al. [29] fabricated GG-NPs containing a galactomannan component that facilitates an uptake by the macrophages having mannose receptors. Upon their application on AD-induced mice, reduced ear thickness and redness, decreased infiltration of eosinophils, macrophages and neutrophils, and repressed serum levels of IgE were evidenced in animals treated with GG-NPs compared to the control groups. These results were attributed to the pronounced permeation of PNPs through the skin and their subsequent retention in the epidermis and the dermis [29].

### 3.4. Skin Cancer

#### 3.4.1. Pathogenesis of Skin Cancer

Skin cancer, which is defined by the uncontrolled growth of cells in the skin tissue [10], is one of the widespread forms of malignancy in humans with about 1.0 million new cases every year [87]. The most common type of skin cancer is keratinocyte cancer which is divided into basal cell carcinoma (BCC) (starts in the epidermal basal layer) and squamous cell carcinoma (SCC) (starts in the upper part of the epidermis), while melanoma, a cancerous tumor of melanocytes [10], occurs rarely [88], it is considered the most aggressive and dangerous form of skin cancer accounting for increased death rates [89]. 

The skin is subjected to harmful substances in the surrounding environment, such as toxic chemicals and ultraviolet (UV) radiation, which may result in the abnormal cells growth or cancer [90]. UV rays and especially UVB is one of the major risk factors for the development of BCC and SCC [91]. The formation of cancerous cells is primarily triggered by the mutation of DNA [92], loss of activity of some tumor suppressor genes such as P53 [93], or the activation of specific oncogene [94]. In addition, human papillomavirus (HPV) infection and immunodeficiency disorders are also risk factors for developing skin cancer [95].

#### 3.4.2. Conventional Treatments for Skin Cancer and Their Limitations

Traditional treatments for skin cancer, which may be used alone or in combination, include surgery, radiotherapy, immunotherapy, and chemotherapy. All these techniques are being conventionally used for the treatment of skin cancer; however, they are also associated with several drawbacks, including a low efficacy or inadequate response, off-target effects [96], cancer resistance [97], scars, and susceptibility to other diseases once the immune system is weakened that limit their clinical significance [98]. 

Chemotherapy, as one of these techniques, is the most employed treatment modality for the treatment of skin cancer. The most commonly used chemotherapeutic drugs such as vinblastine, vincristine, cisplatin, doxorubicin, paclitaxel, 5-flurouracil (5-FU), and bleomycin are primarily administered systemically and due to a lack of selectivity, these drugs may cause serious systemic toxicity, including the gastrointestinal side effects (e.g., nausea vomiting, and diarrhea), hair loss, changes in body weight, severe fatigue, aplastic anemia as a result of bone marrow suppression, leading to a risk of easy bruising or bleeding, and vulnerability to systemic infection [99]. In addition, some chemotherapeutic drugs such as 5-aminolevulinic acid (5-ALA), a prodrug for topical PTD [100], can be administered topically due to its reasonable permeability into former skin layers and thus can show promising results in treating superficial SCC [101]. However, due to lack of deeper penetrability, polarity, an insufficient accumulation into deeper skin layers, and susceptibility to biodegradation due to enzymes in skin tissues, its therapeutic significance is reduced for the treatment of invasive SCC [102,103]. These limitations led the researchers and scientists to search for alterative efficient strategies for the topical management of skin cancer while alleviating the systematic toxicity associated with the administration of chemotherapeutic agents.

#### 3.4.3. PNPs-Based Topical Therapies for Treatment of Skin Cancer

Encapsulation of chemotherapeutic agents into PNPs can improve their physicochemical properties, pharmacokinetics, targeted biodistribution, and anticancer efficacy while reducing their side effects. The majority of PNPs that have been investigated as carriers for the treatment of skin cancer were made up of synthetic or semi-synthetic polymers. For example, Wang et al. [101] investigated PLGA-NPs for the topical delivery of ALA for the treatment of cutaneous SCC. The results suggested that PNPs exhibited a better cell uptake, sustained release, and significantly higher cytotoxicity against SCC cells compared to free ALA [100]. Later on, the same research group also carried out an in vivo study on SCC tumor-bearing mice (male SKH-1 hairless mice) for the evaluation of the delivery potential of ALA-PLGA-NPs for the treatment of skin cancer [104]. They formulated 0.8% ALA or ALA-PLGA-NPs cream and applied them topically to SCC tumor-bearing animals followed by repeated PDT treatment. The results showed that ALA-PLGA-NPs exhibited a significantly higher amount of photosensitizer protoporphyrin IX (PpIX) in SCC compared to free ALA. In addition, ALA-PLGA-NPs-mediated topical PDT showed a better antitumor efficacy in contrast to the same concentration of free ALA. The in situ fluorescence examination showed that the fluorescence intensity in the mouse treated with ALA-PLGA-NPs was stronger and more localized at the tumor site compared to the mouse treated with free ALA, suggesting that ALA-PLGA-NPs displayed selective targeting to SCC [104]. 

Likewise, da Silva et al. [105] developed PpIX-loaded PLGA-NPs for topical delivery. The in vitro release study showed a sustained release of PpIX from PLGA-NPs up to 10 days. In addition, an ex vivo skin retention study showed a 23-fold higher retention of PpIX-PLGA-NPs in the SC and 10-fold higher in the epidermal and dermal layers compared to free PpIX. These results were also validated in hairless mice after a topical application of PpIX-PLGA-NPs where an approximately three-fold higher retention in the SC and two-fold higher retention in the epidermis and dermis were observed in case of PpIX-PLGA-NPs compared to free PpIX. These results indicate that PNPs are suitable for the PDT of skin cancer [105].

Rata et al. [43] fabricated 5-FU-loaded CS and poly(N-vinylpyrrolidone-alt-itaconic anhydride) nanocapsules that were then incorporated into a topical gel formulation formed of sodium alginate and HA. The formulated hybrid PNPs-incorporated gel was tested for drug permeation through the chicken skin membrane. A significantly higher permeability of 5-FU was evident in case of PNPs compared to free 5-FU. Moreover, the hemolysis assay, skin irritation test, and cytotoxicity assay carried out on human basal carcinoma cell line (TE 354.T) proved a good compatibility of PNPs with blood and non-irritant to the skin. An in vitro cytotoxicity study showed a significant decline in the viability of cancer cells treated with PNPs compared to free 5-FU [43]. 

Similarly, Sabitha et al. [44] fabricated 5-FU-loaded chitin (a highly abundant natural polymer) nanogels (FCNGs) for the topical management of skin cancer. A pH-responsive swelling with the subsequent release of 5-FU was evidenced from FCNGs. At a concentration range of 0.4–2.0 mg/mL, FCNGs exhibited a reasonable cytotoxicity against human melanoma cells (A375) while exerting minimal toxicity on normal human dermal fibroblasts (HDF). FCNGs were found to be safe with no signs of blood hemolysis (very low hemolytic ratio) even at higher concentrations, which indicates their good biocompatibility. FCNGs and free 5-FU exhibited a similar steady state flux; however, a 4–5 times higher retention of 5-FU in deeper skin layers was evident from FCNGs compared to free 5-FU. The histological evaluation showed the fluidization of the epidermal horny layer after an interaction with cationic chitin without causing any inflammation, which was anticipated to be the possible mechanism by which FCNGs promote the skin permeation of 5-FU and thus could be a promising treatment for skin cancer [44].

### 3.5. Skin Infections

#### 3.5.1. Pathogenesis of Skin Infection

Skin is highly susceptible to microbial infection since it is the largest organ of the body and serves as the first-line barrier against invaded microorganisms. Compromised skin tissues are highly vulnerable to colonization by invading pathogens which may form a biofilm on the surface of the skin and soft tissues [49]. There are several types of skin infections depending upon the type of causative microorganism. For example, skin bacterial infections are caused by *Escherichia coli*, *Staphylococcus aureus,* or *Pseudomonas aeruginosa* [106,107], fungal infections are caused by *Candida* or *Aspergillus* species, [108], parasitic or protozoal infections are caused by *Leishmania* species [109], and viral infections are caused by *herpes virus* or *pox virus* [110].

The penetration of pathogenic microorganisms into the skin may damage skin tissues by triggering the inflammatory and immune responses. Pathogens may penetrate skin tissues through the damaged skin such as cuts, scratches, bites, puncture produced by the needles, or pre-existing skin conditions. Skin infection occurs in three major stages: the adherence of microbe to host epithelial cells, invading to skin tissue by bypassing the SC, and releasing the toxins. Some exotoxins, for instance, lipases, which are released by *S. aureus* induce the digestion of skin fatty acids and penetrate to deeper skin layers. The other part of the infection process is inflammation that is generally a protective body response to tissue damage. As a result, an increase in blood flow and the extravasation of leukocytes from the blood vasculature to the site of infection takes place. The main manifestations of inflammation include pain, erythema, warmth, and edema. In the case of a severe infection, additional systemic signs and symptoms such as fever, tachycardia, or hypotension may occur which result from changes in vascular resistance and thermoregulation induced by cytokines (inflammatory mediators) [111].

#### 3.5.2. Conventional Treatments for Skin Infections and Limitations

Skin and soft tissues infections are usually treated with systemic (oral or intravenous) antimicrobial agents since the topical treatments (e.g., creams, ointments, gels, or sprays) alone are not adequate or effective most of the times. In case of mild skin lesions, oral therapy can be used in outpatients, while moderate-to-severe lesions may need outpatient intravenous therapy or hospitalization [111].

Most commonly used antibacterial agents for skin infections include penicillins (e.g., amoxicillin, cloxacillin, etc.), cephalosporins (e.g., cephalexin, ceftriaxone, etc.), fluoroquinolones (e.g., ciprofloxacin), and lincomycin (e.g., clindamycin) [111]. The majority of antibiotics produce almost similar side effects such as stomach pain, nausea, vomiting, diarrhea, and headache that are usually mild and self-limiting [112]; however, some antibiotics may produce severe adverse effects. For example, chloramphenicol, which is a broad-spectrum antibiotic, is usually administered systemically to treat skin infections; however, the systemic administration of this drug is associated with severe adverse effects such as bone marrow toxicity [113]. Additionally, fluoroquinolones, which are given orally or intravenously, may induce infrequent but very serious side effects such as the prolongation of QT-interval-causing abnormal heart rhythms [114].

Anti-parasitic drugs that are mainly used to treat leishmaniasis (a disease caused by the parasites of the genus *Leishmania*) include antimonials, alkylphospholipids (e.g., miltefosine), aminoglycosides (e.g., paromomycin), and polyenes (e.g., amphotericin B which is used as antifungal as well). These drugs have several drawbacks such as high costs, a low efficacy, variable cure rates, and toxicity, which reduce their clinical significance and patient compliance [115]. For instance, amphotericin B, which is administered intravenously, is associated with nephrotoxicity [116], and aminoglycosides may induce ototoxicity, which are irreversible side effects [112].

Some antibiotics are also available as topical preparations as creams or ointments such as acyclovir (antiviral drug), but they can cause some topical side effects such as skin irritation, itching, burning, mild pain, and dryness [117]. Due to all these limitations associated with conventional medications, researchers are in contestant investigations to find new drugs or formulations that are safe, effective, affordable, and preferably applied topically to minimize systemic adverse events.

#### 3.5.3. PNPs-Based Topical Therapies for Treatment of Skin Infections

To mitigate various challenges associated with a topical administration as well as to improve the therapeutic efficacy of anti-infective agents, many researchers attempted the nanoencapsulation of anti-infective agents into PNPs. To evaluate the efficacy of PNPs in the treatment of a fungal infection, Mumtaz et al. [45] synthesized CS-NPs loaded with voriconazole (VRC) and formulated in the form of film-forming topical spray (FFS). Ex vivo skin permeation and retention studies carried out on dermatome mice skin indicated a significantly higher permeation and deposition of VRC-CS-NPs-FFS in the SC, epidermis, and dermis compared to conventional VRC suspension. In addition, VRC-CS-NPs-FFS resulted in a six-fold greater inhibition against *C. albicans* and *A. flavus* compared to VRC suspension. The local irritation study carried out on albino rabbits showed no signs of skin irritation and toxicity compared to free VRC suspension. These results indicate that the encapsulation of VRC into PNPs not only improves the anti-fungal efficacy but also the safety of VRC [45].

Likewise, Pervaiz et al. [30] developed terbinafine (TB)-loaded CS/XG-NPs that were subsequently incorporated into Carbopol gel to treat *C. albicans*-associated skin infection in albino rats. The complete eradication of fungal colonies was evident following 21 days of treatment with TB-loaded PNPs-incorporated topical gel compared to commercial cream and gel. The superior antifungal activity of TB-loaded PNPs was attributed to a higher skin permeation and retention (approximately 78%) into the skin layers compared to commercial cream and gel formulations [30]. Similarly, Ho et al. [46] encapsulated itraconazole (ITZ) into ethylcellulose (EC)-NPs and then incorporated them into a topical gel formulation with semisolid consistency. A comparative analysis on rat epidermis revealed an approximately 20-fold higher permeability and 7-fold higher skin retention of ITZ-EC-NPs released from the topically applied gel compared to ITZ suspension gel. Furthermore, ITZ-EC-NPs gel exhibited a significantly higher fungal inhibition against *C. albicans* and *A. niger* compared to ITZ dispersion [46].

Amphotericin B (AmB) is a potent fungicidal agent for the treatment of a wide range of fungal infections; however, due to its poor aqueous solubility, it is limited to an intravenous administration. In search for the possible delivery of AmB through the topical route, Riezk et al. [47] developed AmB-loaded CS-TPP (AmB-CS-TPP) or CS-dextran sulphate (AmB-CS-Dex) PNPs. The in vitro characterization of developed PNPs revealed nanoscaled dimensions and the sustained release of AmB from both types of PNPs but an approximately 1.5-fold greater release from AmB-CS-TPP compared to AmB-CS-Dex. The evaluation of the antifungal activity against *Leishmania* indicated that all formulations exhibited reasonable antifungal efficacy; however, emulsion formulation incorporated with AmB-PNPs exhibited the highest antifungal efficacy compared to free AmB and AmBisome^®^. In addition, AmB-loaded PNPs exhibited a better biocompatibility against red blood cells (RBCs) and human squamous carcinoma (KB) cells compared to free AmB and AmBisome^®^ [47].

Acyclovir (ACR) is considered a gold standard drug for the treatment of skin viral infections caused by herpes simplex or varicella-zoster virus; however, its poor dermal permeability and low aqueous solubility limit its topical application. To overcome these challenges, Abd-Elsalam et al. [48] fabricated lipid-coated CS nanocomplexes (LCNCs) for the topical delivery of acyclovir (ACR). To further improve its dermatokinetics, ACR-LCNCs were further functionalized with span-80 and TPGS (D-α-tocopheryl polyethylene glycol succinate). Span-80 is commonly used as a surfactant in various topical formulations to enhance dermal permeation by manipulating the permeation coefficient of a drug as well as temporary disrupting the SC. TPGS is also an FDA-approved hydrophilic surfactant commonly used as an adjuvant ingredient (emulsifier, solubilizer, penetration enhancer, bioadhesive, etc.) in topical formulations. The results suggested a significantly higher permeation and skin retention of ACR from span-80/TPGS-functionalized ACR-LCNCs compared to unfunctionalized ACR-LCNCs and free ACR dispersion. Furthermore, tolerance and safety studies carried out in healthy Wistar rats showed no signs of erythema and with normal histological features of the skin treated with drug-free LCNCs and span-80/TPGS-functionalized ACR-LCNCs, which validate the biocompatibility and safety of PNPs [48].

Another innovative topical nanocomposite hydrogel formulation was developed by Patarroyo et al. [49] to counter antimicrobial resistance. Antimicrobial resistance is becoming worrisome due to the excessive and inappropriate use of antibiotics. To overcome this concern as well as for the better treatment of microbial infections, a hybrid nanocomposite hydrogel composed of a polymeric network of gelatin–polyvinyl alcohol–hyaluronic acid that is incorporated with the nanoconjugates of graphene oxide (GO)/silver (Ag)-NPs. GO-Ag-NPs have been extensively investigated for superior antimicrobial activity compared to Ag-NPs. The developed nanocomposite hydrogel showed a good biocompatibility with more than 80% cell viability (Vero cells) at low-to-moderate concentrations, minimal hemolytic effect (<5%), and moderate platelet aggregating capacity (35–45%). Compared to commercial formulations, GO-Ag-NPs nanocomposite hydrogel exhibited a 100% inhibition of *S. aureus* and *E. coli* growth at 20 μg/mL. These findings clearly evidenced the biomedical potential of polymeric nanocomposite hydrogel for their topical application for the treatment of microbial infections and infected wounds [49].

### 3.6. Skin Wounds

#### 3.6.1. Pathogenesis of Skin Wounds

Skin wound is a very common type of injury to skin tissues. There are many causes that may result in cutaneous wounds including burns and cuts, and some of them can be life-threatening, such as chronic diabetic wounds and wounds associated with peripheral arterial diseases that fail to close and require a longer time to heal [118]. The natural wound healing process consists of a specific cellular and molecular mechanism involving four distinct phases that include: (1) hemostasis, (2) inflammation, (3) accumulation of extracellular matrix, proliferation, and granulation tissue formation, and (4) remodeling and formation of scar (Figure 6) [119]. Many receptors and growth factors play important roles in the normal wound healing process. Wounds can be classified according to the appearance, damage rate of skin tissues, and the cause of injury [120].

The normal wound healing process takes about 4–6 weeks; however, chronic wounds such as diabetic wounds or foot ulcers take a relatively longer time to heal. This delay in the wound healing process can be due to a poor control of the blood glucose level, bacterial colonization, ischemia, hypoxia, altered cellular response, or defects in collagen synthesis and deposition. Additionally, a reduction in plasma fibronectin, chemokines, and pro-angiogenic growth factors may delay an inflammatory response and deactivates fibroblast proliferation, resulting in apoptosis in the wound bed [121].

#### 3.6.2. Conventional Therapies for Acute-to-Chronic Wounds and Limitations

Conventionally, the most commonly available options for the treatment of non-infected skin wounds include topical drugs, gels, creams, ointments, and occlusive dressings that retain moisture, compression bandages, surgical debridement for the removal of damaged tissue or foreign objects from wounds, and wound closure using vacuum [122]. However, in the case of an infected wound, antimicrobial agents are critically important. For example, silver dressing, moxifloxacin (antibiotic) [123], or antiseptic solutions such as povidone-iodine, hydrogen peroxide, and Dakin’s solution (sodium hypochlorite) are commonly used on infected wounds to reduce microbial biofilm [124]. These antimicrobial treatments are also associated with toxicity of skin tissues. 

Although conventional formulations display a reasonable wound healing efficacy; however, they are also associated with several drawbacks such as poor residential time at the wounded site. This leads to low dosing, requires more frequent applications, poor drug penetration into the necrotic tissues in case of chronic wounds, and low drug retention into skin layers for a local anti-inflammatory effect and tissue regeneration [125]. Furthermore, the removal or replacement of conventional dressings usually causes the stripping off the newly formed upper skin layer (epidermis), which results in delaying wound healing [126]. Rapid wound healing is much more desirable to prevent risks of infection and other serious complications; therefore, it is essential to find new functional delivery systems with a better potential to accelerate the wound healing process with minimal local and systemic toxicity.

#### 3.6.3. PNPs-Based Topical Therapies for Wound Healing

Hussain and colleagues [28] developed HA-functionalized CS-NPs co-loaded with CUR and resveratrol (REV) for the topical management of burn wounds. The in vitro release kinetic study revealed a biphasic release pattern from HA-CUR-REV-CS-NPs with an initial fast release of CUR and REV (first 8 h) followed by the continuous sustained release. Due to the hydrophobicity of CUR, it showed a relatively slower release than the REV. Drugs were released from the PNPs via non-Fickian diffusion (through diffusion and erosion of the polymeric matrix). Authors anticipated that the functionalization of PNPs with HA would reduce the frequency of a topical application, prolong the localization of drugs at the target site, and improve the wound healing efficacy for the treatment of burn wounds [28]. 

Similar research also validated the pharmaceutical viability of PNPs for the topical co-delivery of quercetin (QUE) and CUR [50]. The prepared HA-CUR-QUE-CS-NPs exhibited a triphasic release pattern and Fickian diffusion with sustained release kinetics. Drugs permeation and retention into rat skin (ex vivo) were assessed and the results showed that the permeation efficiency of HA-functionalized PNPs was approximately three times higher than non-functionalized PNPs. Similarly, HA-functionalized PNPs showed a higher retention of drugs in the epidermis and dermis by 1.8 times and 2 times, respectively compared to bare PNPs. The developed PNPs-based topical cream formulation (nanocream) was also tested for its efficacy for burn wound healing using Wistar rats. Interestingly, throughout the treatment period, wound healing was more efficacious in the animal group treated with HA-CUR-QUE-CS-NPs (wound closure rate of 98% on day 28) compared to CUR-QUE-CS-NPs group (77%), control group (28%), and untreated group (11%). Moreover, histological analyses exhibited the low infiltration of inflammatory cells, the deposition of collagen, and re-epithelization [50].

Another novel topical formulation was developed by Shafique and colleagues for the management of infected cutaneous wounds [51]. They used a hydrogel, a crosslinked three-dimensional network of hydrophilic polymers that can absorb water or biological fluids, in the form of thin films known as a hydrogel membrane that can be used as a vehicle for free drugs or drug-loaded PNPs. The soft, swollen hydrogel mimics animal tissues and it is suitable for an application on wounds [127]. PNPs that encapsulate drugs need a base or vehicle to be embedded within them, such as these type of hydrogel membranes. Hence, they prepared CS-NPs loaded with cefepime (a fourth-generation cephalosporin antibiotic) and then embedded into the hydrogel membrane formed of HA, polyvinyl alcohol, and pullulan (a polysaccharide polymer). The developed hydrogel membrane showed a good swelling capacity and efficient release of cefepime (88%) in a sustained manner. 

It is important to have a humid environment and prevent the dehydration of a wound, so calculating the water vapor transmission rate (WVTR) of a wound dressing is required. Furthermore, oxygen permeability through dressing is essential for cell growth and to promote the wound healing process. The calculated WVTR was between 2000 and 2500 g/m^2^/day, and oxygen permeability was between 7 and 14 mg/L; both values lie within the ideal dressing range. The cefepime-loaded hydrogel membrane exhibited a higher zone of inhibition against different Gram-positive and Gram-negative bacteria, including the *S. aureus*, *E. coli*, and *P. aeruginosa* that could infect the skin and soft tissues on the wounded area. In addition, the hydrogel membrane exhibited no cytotoxicity against HT1080 (human fibrosarcoma cell line). Furthermore, in vivo tests were performed on Sprague Dawley rats and an accelerated wound healing was observed with a wound closure rate of 80% on day 7 and 100% on day 14 in animals treated with cefepime-loaded PNPs-embedded hydrogel membranes. This formulation helped in a faster recovery since HA can facilitate fibroblasts proliferation and migration in addition to keeping the wound bed moist by preventing the water evaporation, which is highly desirable for faster wound healing. 

Additionally, pullulan, as an important ingredient of the developed hydrogel membrane, acted as a source of energy for epithelial cells and helped them in regenerating the skin through the consumption of glucose [51]. The same research group had also previously engineered a thermosensitive hydrogel membrane for the topical delivery of amikacin, which is an aminoglycoside antibiotic with a strong antimicrobial efficacy against a wide range of skin infections, including the infected cutaneous wounds [128]. In this study, they used sodium alginate and other materials including poloxamer 407, pluronic F-127, and polyvinyl alcohol for making the thermosensitive hydrogel membrane. An in vitro characterization showed that the developed hydrogel membrane exhibited good swelling characteristics, porosity, mechanical properties, and sustained release behavior. In comparison to free drug, the hydrogel membrane displayed a significantly higher zone of inhibition against *S. aureus* and *P. aregnosa*. The in vivo testing of the hydrogel membrane showed faster wound healing and the greater granulation of tissue formation and re-epithelization. The superior wound healing efficacy of developed natural polymeric hydrogel membrane was attributed to strong tissue regeneration, re-epithelization, anti-inflammatory, wound bed moistening, and the antimicrobial properties of sodium alginate, poloxamer 407, pluronic F-127, and polyvinyl alcohol [128].

Curcumin (CUR) is a natural polyphenol with excellent anti-inflammatory, antimicrobial, anticancer, antioxidant, and tissue regenerating properties; however, its inherent physicochemical properties such as a poor solubility, low bioavailability, and photo-degradability restrict its clinical translation. In an attempt to improve its physicochemical properties and chemical stability, Nguyen et al. [129] developed oligo-chitosan (OCHI)-based nanoplexes for the topical delivery of CUR for the treatment of wounds. The developed nanoplexes (CUR-OCHI) possessed nanoscaled dimensions, a good encapsulation efficiency (>90%), and high zeta potential. In addition to having a good safety profile (no signs of toxicity against human keratinocytes), fabricated PNPs exhibited a good wound healing rate with >90% wound closure achieved within 7 days of treatment in contrast to free CUR [129]. 

Likewise, Mirnejad et al. [130] synthesized CS-NPs, aiming to improve the topical delivery and wound healing efficacy of CUR against infected cutaneous wounds. Interestingly, fabricated CUR-PNPs displayed a very good antimicrobial efficacy (in vitro) against *S. aureus* and *P. aeruginosa*. With this excellent antimicrobial efficacy, the authors anticipated that CUR-PNPs could be a potential wound healing modality for infected cutaneous wounds [130]. Similar findings have also been reported by other studies [131,132]. 

Krausz et al. [131] fabricated PNPs for the topical delivery of CUR and evaluated their wound healing potential against the infected burn wounds. The developed CUR-PNPs exhibited desired physicochemical properties, smooth spherical morphology, a good stability, and sustained release kinetics. In addition to exhibiting a high degree of safety against PAM212 cells (mouse keratinocytes), the developed CUR-PNPs showed excellent antibacterial efficacy against methicillin-resistant *S. aureus* (MRSA) and *P. aeruginosa*. The in vivo evaluation of wounded animals showed a great improvement of the wound healing rate with an enhanced granulation tissue formation and re-epithelization, and good antimicrobial and anti-inflammatory efficacies compared to free CUR [131].

Another research has recently engineered novel natural lignin-based PNPs (LNPs) for improving the physicochemical properties and wound healing efficacy of CUR [132]. The encapsulation of CUR into PNPs resulted in a significant improvement in the aqueous solubility, chemical stability, and antibacterial efficacy against *S. aureus*. An in vitro wound healing assay (scratch assay) demonstrated the good migration and proliferation of fibroblasts, which is a prerequisite to granulation tissue formation and re-epithelization in the wound healing process. An animal study (Wistar rat) also showed a superior wound healing efficacy of CUR-PNPs on excisional cutaneous wounds with complete wound closure within 12 days of treatment (Figure 7). These results validated lignin as a potential natural polymer for fabricating PNPs for the topical delivery of bioactives to promote wound healing efficacy [132].

Another novel hybrid scaffold embedded with CUR-loaded CS-NPs was developed by Karri et al. [133]. Following the synthesis, CUR-loaded CS-NPs were incorporated into a collagen-alginate scaffold for the topical management of cutaneous wounds. The developed CUR-PNPs exhibited a nanoscaled dimension, good stability, high encapsulation efficiency, and sustained release profile along with improving the physicochemical properties and chemical stability of CUR. The in vivo testing on animals showed that cutaneous wounds treated with the CUR-PNPs-scaffold showed a faster wound closure with a higher granulation tissue formation and re-epithelization compared to wounds treated with free CUR or a plain scaffold [133]. The findings from all these studies validated that PNPs possess a massive potential as a carrier for the topical delivery of a wide range of therapeutics for the treatment of acute-to-chronic wounds.

### 3.7. Acne

#### 3.7.1. Pathogenesis of Acne

Acne is a chronic inflammatory skin disorder that occurs commonly in teenagers and can be associated with various factors such as hormone imbalance, puberty, age, and stress [134]. The exact cause of acne is not fully established; however, it is believed to be due to *Propionibacterium acnes* (*P. acnes*) found in the normal microflora in follicular units that stimulates a pro-inflammatory cytokine release, especially ILs, which induce immune responses. Follicular keratinocyte hyper-proliferation concomitant with an increased sebum production mediated by androgen exacerbate the condition by blocking the skin pores [135]. Moreover, complex inflammatory pathways involving the innate and acquired immunity also believed to contribute to acne formation [136].

#### 3.7.2. Conventional Treatments for Acne and Limitations

Treatment for acne varies according to the stage of the disease. In mild-to-moderate cases, a topical treatment with retinoids or antimicrobial agents or a combination of both is the gold standard therapy [137]. In addition to their anti-inflammatory properties, topical retinoids prevent follicular blockage by accelerating the shedding of corneocytes, making way for new cell growth underneath. Common topical retinoids include tretinoin, adapalene, and tazarotene [138]. On the other hand, benzoyl peroxide (BP) also possesses keratinolytic and anti-inflammatory properties along with bactericidal properties against *P. acnes*, which is mainly attributed to the generation of reactive oxygen species (ROS) [139]. It can also be used in combination with other antibiotics such as erythromycin or clindamycin to mitigate the development of *P. acnes* resistance [140]. The most common side effects associated with the topical use of retinoids and BP include skin irritation, erythema, and scaling [141]. Other over-the-counter topical agents that are frequently used for acne treatment include salicylic acid, resorcinol, sulfur, zinc, and aluminum chloride [142].

When topical therapy is ineffective or in the case of moderate-to-severe acne, a systemic treatment can be advised. Oral antibiotics in combination with topical retinoids and BP can be used to reduce risks of possible bacterial resistance [143]. Tetracyclines including minocycline, doxycycline, and tetracycline are considered the first line treatment, except when they are contradicted such as during pregnancy, in children <9 years, or in the case of an allergy; in such cases, erythromycin can be used as an alternative [144]. In addition, hormonal therapy can be used to reduce the severity of acne. Oral contraceptives containing estrogen can be effective by reducing the levels of testosterone [145]. Oral spironolactone (an antiandrogen) has also shown a moderate efficacy in acne treatment; however, its use is limited by numerous side effects such as hyperkalemia, menstrual irregularities [146], and neurological abnormalities [147]. In treating resistant acne, the use of isotretinoin (IST) can be considered. It reduces sebum production, acne scarring, and *P. acnes* colonization [148]. Because of its potential serious side effects, blood tests including CBC (complete blood count), blood lipid, liver enzymes, and blood glucose levels should be done before starting the treatment and at monthly bases thereafter [149,150].

#### 3.7.3. PNPs-Based Topical Therapies for Treatment of Acne

Due to the involvement of pilosebaceous units (PSU) (comprised of hair shaft, hair follicle, sebaceous gland, and erector pili muscle) in the pathogenesis of acne, many research efforts have been directed toward targeting these structures using the PNPs [151]. For example, Tolentino et al. [27] developed CS-NPs and HA-NPs for the topical delivery of clindamycin. Both PNPs demonstrated a significantly higher deposition of clindamycin into PSU (53 ± 20% from CS-NPs and 77 ± 9% from HA-NPs) compared to commercial formulations [27]. Similarly, Ogunjimi et al. [52] developed *Delonix* (DLX) (a galactomannan polysaccharide)-based PNPs for the topical delivery of isotretinoin (IST) and evaluated them for the treatment of acne. A skin distribution study in pig ear skin showed a three-fold increase in the accumulation of IST in the hair follicles in case of IST-DLX-NPs compared to IST solution. In addition, testing on AMJ-2 macrophages to assess the inflammatory response showed a significant repression in the expression of pro-inflammatory cytokines such as IL-6, TNF-α, and IL-10 compared to control groups and IST solution. Furthermore, an in vivo skin irritancy study carried out on Wistar rats revealed the lowest UVA-induced irritation in the skin treated with PNPs compared to IST solution [52].

In another study conducted by Kandekar et al. [53], adapalene (ADA) was loaded into PNPs composed of TPGS. The in vitro finite dose experiment conducted on porcine and human skin demonstrated an approximately 2- and 10-fold higher deposition of ADA into PSU from PNPs dispersion and PNPs gel in contrast to marketed ADA gel and cream formulations. Later on, a follicular delivery study in human skin (on intact PSU) demonstrated an approximately 4.5- and 3.3-fold higher ADA accumulation into PSU from PNPs dispersion and PNPs gel compared to the PSU-free skin biopsies [53].

*P. acnes* play a critical role in the pathogenesis of acne as well as in developing antimicrobial resistance against the most used antibiotics. To counter this problem, Friedman et al. [54] developed CS/ALG PNPs and evaluated its antibacterial efficacy against P. acnes. Antimicrobial assay demonstrated a dose-dependent bactericidal effect of plain CS/ALG-PNPs against P. acnes and it was attributed to the disruption of the bacterial cell membrane. The anti-inflammatory activity in human monocytes and keratinocytes was evidenced due to the reduction in IL-12 and IL-6 production [54]. Later on, they encapsulated BP into CS/ALG-PNPs and tested against P. acnes, where a dose-dependent antimicrobial effect was observed with no signs of serious toxicity to eukaryotic cells [54].

Other PNPs such as PLGA-NPs have also been studied for the topical delivery of thymol (TH) [55] and its antimicrobial efficacy against *S. epidermidis* and *C. acnes* (in vitro and ex vivo). A significant inhibition against *C. acnes* with minimal effect on *S. epidermidis* were noticed, which would help in preserving the normal microbiota during acne treatment. The developed PNPs also displayed a good antioxidant activity with no cytotoxicity in HaCaT cells (human keratinocytes) at different concentrations up to 50 µg/mL. The permeation studies conducted on healthy and damaged abdominal human skin showed a significantly higher TH permeation in the damaged skin compared to normal skin, and the permeation and amount of TH retained into various skin layers was more pronounced in case of TH-loaded PNPs compared to free drug [55].

Likewise, Kahraman et al. [56] reported the fabrication of BP into PNPs using Pluronic^®^ F127. In vitro skin penetration studies in porcine skin resulted in a three-fold higher accumulation of BP in the skin compared to the commercial gel. This could be explained in terms of the enhanced thermodynamic activity of drugs inside the carrier, resulting in an increased partitioning from PNPs to the skin. Moreover, the ultra-fine particle size of PNPs, which improves the contact area with the skin, also contribute to an enhanced permeation and retention into various skin layers. They also ensured the safety profile of PNPs against the mouse embryonic fibroblasts demonstrated by an approximately 7.6 times higher IC_50_ compared with sodium dodecyl sulphate (SDS), which is a severe irritant of the skin [56]. These findings clearly evidenced the great potential of PNPs for the follicular delivery of bioactive for the treatment of acne. A summary of other PNPs prepared from natural polymers and used for the treatment of topical skin diseases has been presented in Table 1.

## 4. Conclusions and Summary

Topical skin diseases offer substantial challenges to current conventional therapeutic modalities due to the lack of a sufficient amount of drug(s) which are reaching and retained into skin tissues following their administration through different routes. Moreover, the off-target accumulation of drug(s) administered through different routes results in several unwanted systemic adverse events.

To mitigate all these issues, the topical route is considered the most promising port for drug delivery; however, the topical route offers substantial barriers to drugs’ penetration into deeper skin tissues due to the foremost layer of the skin, the stratum corneum (SC). To overcome this barrier, numerous passive and active delivery techniques have been utilized, such as iontophoresis, electroporation, phonophoresis, radiofrequency, laser ablation, microneedles, and chemical penetration enhancers; however, all these techniques are also associated with many side effects, which restrict their clinical acceptability. Alternatively, nanotechnology-aided interventions have been adapted and extensively investigated for the topical administration of a wide variety of therapeutics.

In this review, we have mainly focused on the biopharmaceutical and therapeutic viability of PNPs prepared from natural polymers for the treatment of various skin diseases such as psoriasis, atopic dermatitis, skin infections (bacterial, viral, fungal, etc.), skin cancer, acute-to-chronic wounds, and acne.

A wide variety of natural polymers have been used for the fabrication of PNPs, alone or in combination (polymeric blends), depending upon the desired characteristic features. Owing to the flexibility of varying the type, molecular weight, and concentration of polymer(s), PNPs are extremely tunable nanocarriers that can be optimized for versatile properties. For example, the particle size, zeta potential, polydispersity index, encapsulation efficiency, lading capacity, morphology, %yield, release kinetics, permeation efficiency, pharmacokinetics, and specific interaction with biological targets can be optimized by varying multiple process and formulation parameters during the fabrication process of PNPs.

PNPs have a great potential for the topical delivery of a wide range of pharmacological agents through improving their physicochemical properties, avoiding of premature degradation, sustainable release, optimizing the permeation through the SC, and enhancing their retention into various skin layers (i.e., the SC, epidermis, and dermis). Following their topical application, PNPs may follow intercellular, intracellular, or trans-follicular routes for permeability through the SC. Thereafter, PNPs reside into skin tissues and act as a drug reservoir for prolonging a sustained release and local therapeutic effects. The prolonged localization of drug-loaded PNPs into skin tissues improves the therapeutic outcomes of drugs while minimizing their systemic bioavailability and associated toxicity.

Recently, the functionalization of PNPs has been well-studied as a new adaptation in the design of PNPs to further improve their dermatokinetics and therapeutic efficacy. For example, PEGylation, surface coating with biomolecules, conjugation with targeting ligands, and pH- or thermo-responsive delivery have significantly improved their therapeutic efficacy against various topical skin diseases compared with un-functionalized PNPs. A plethora of in vitro, ex vivo, and in vivo studies have evidenced the superior therapeutic efficacy of PNPs for the treatment of topical skin diseases; however, data on clinical evaluation are scarce. Owing to their exceptional potential, researchers and scientists should further explore the clinical significance of applying PNPs via conducting clinical trials that could offer a potential future perspective for the treatment of challenging topical skin diseases.

## Figures and Tables

**Figure 1 pharmaceutics-15-00657-f001:**
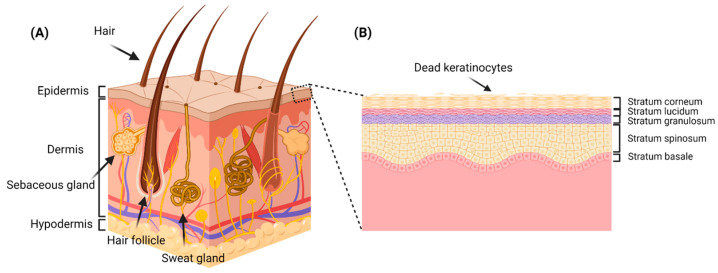
(**A**) Skin structure showing the three distinct layers of epidermis, dermis, and hypodermis, and skin appendages including hair follicle and shaft, sebaceous gland, sweat gland, and arrector pili muscle; and (**B**) epidermis structure showing the five distinct layers of stratum corneum, stratum lucidum, stratum granulosum, stratum spinosum, and stratum basale. Created with BioRender.com (accessed on 4 February 2023).

**Figure 2 pharmaceutics-15-00657-f002:**
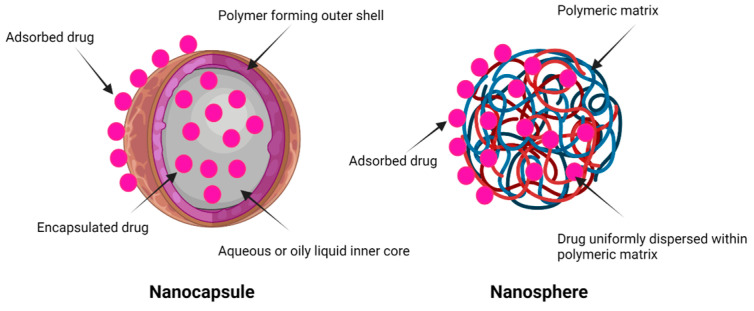
Morphological difference between the nanospheres and nanocapsules. Created with BioRender.com (accessed on 4 February 2023).

**Figure 3 pharmaceutics-15-00657-f003:**
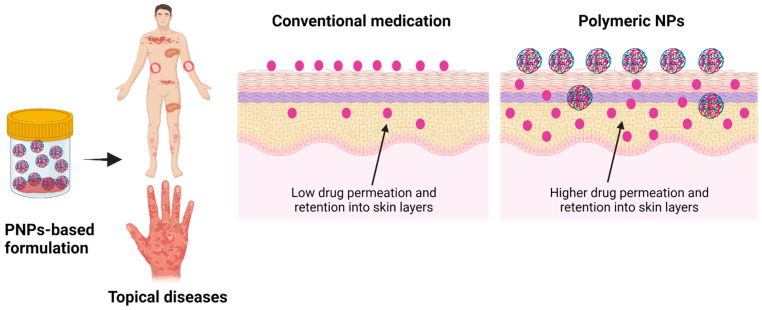
Increased permeation efficiency of PNPs through the skin and retention into various skin layers. Created with BioRender.com (accessed on 4 February 2023).

**Figure 4 pharmaceutics-15-00657-f004:**
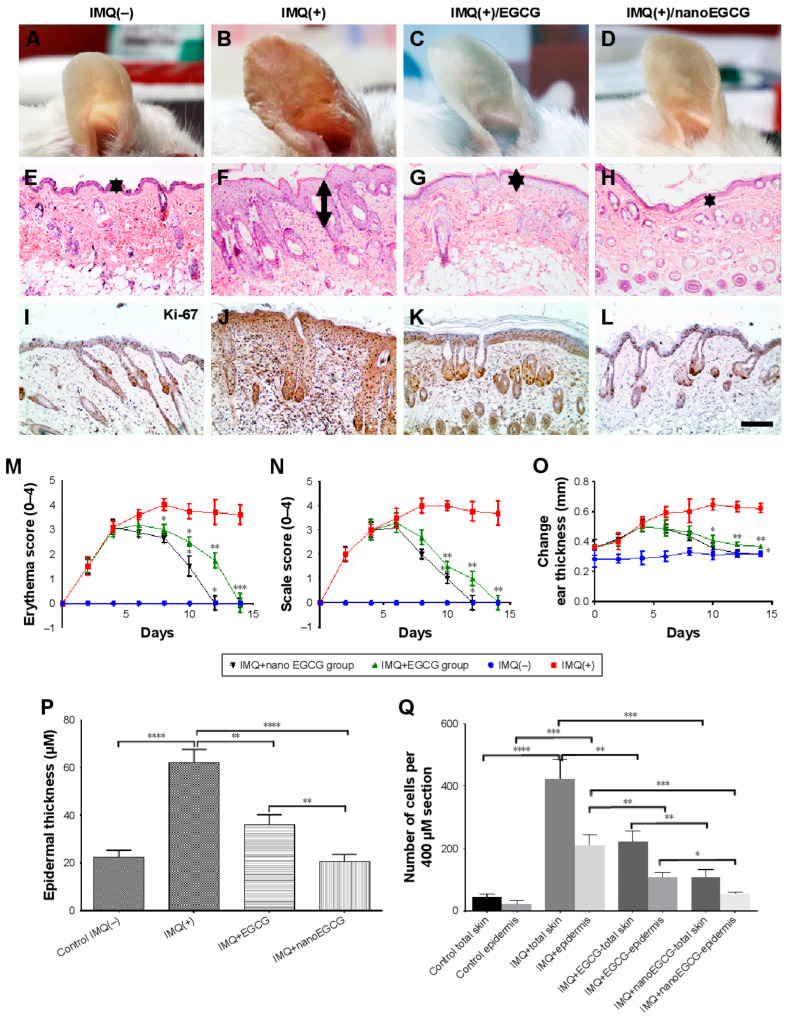
Comparison between topical free EGCG and nanoEGCG in terms of clinical efficacy and histological evaluation in IMQ-induced mice. (**A**–**D**) inflammation severity in mice ears. (**E**–**H**) H&E-stained sections of skin. (**I**–**L**) Ki67-stained sections of skin. (**M**–**Q**) quantitative evaluation of psoriatic features. * *p* < 0.05, ** *p* < 0.01, *** *p* < 0.001, **** *p* < 0.0001. Adapted from [33], Dove Medical Press, 2018.

**Figure 5 pharmaceutics-15-00657-f005:**
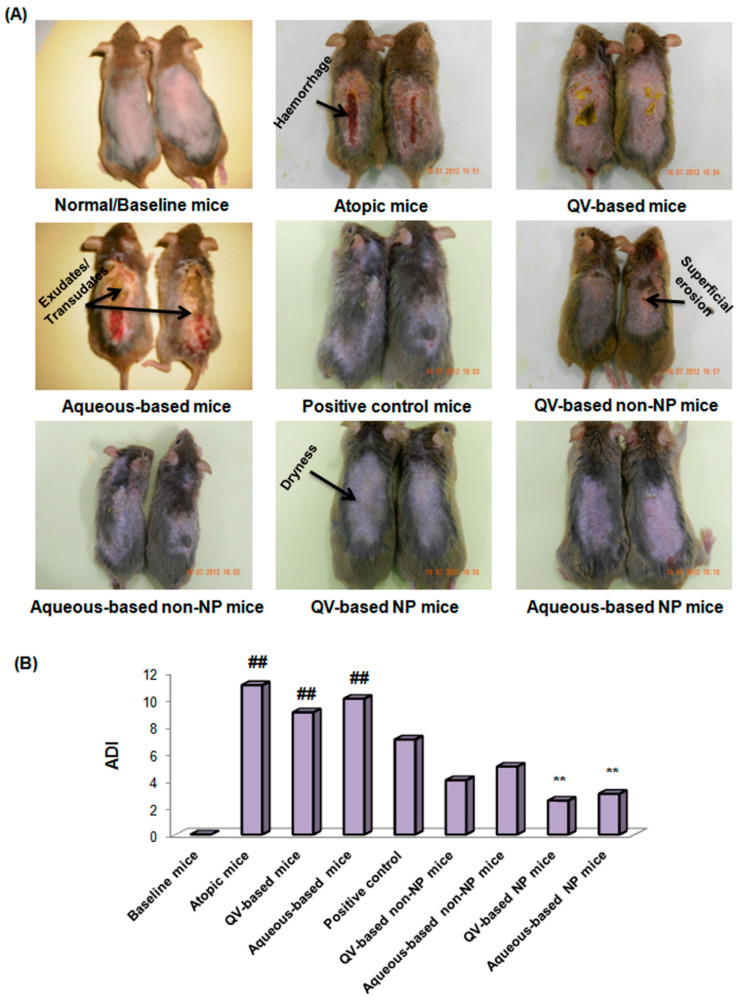
Superior anti-AD efficacy of PNPs-based formulation in NC/Nga mice compared to other treatment groups. ** *p* < 0.001, ^##^
*p* < 0.005. Adapted from [41], PLOS, 2014. Digital images (**A**) and ADI (**B**) of untreated and treated AD-induced NC/Nga mice.

**Figure 6 pharmaceutics-15-00657-f006:**
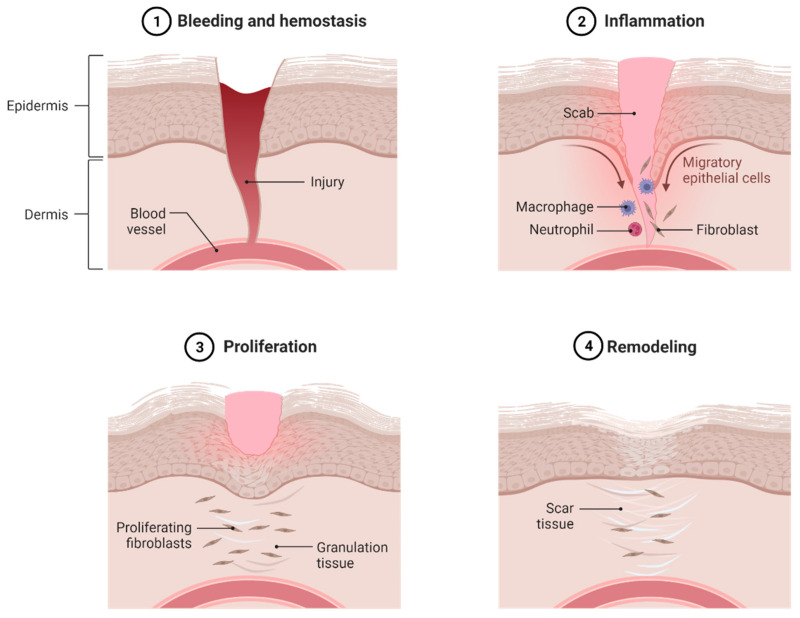
Stages of normal wound healing process. Created with BioRender.com (accessed on 4 February 2023).

**Figure 7 pharmaceutics-15-00657-f007:**
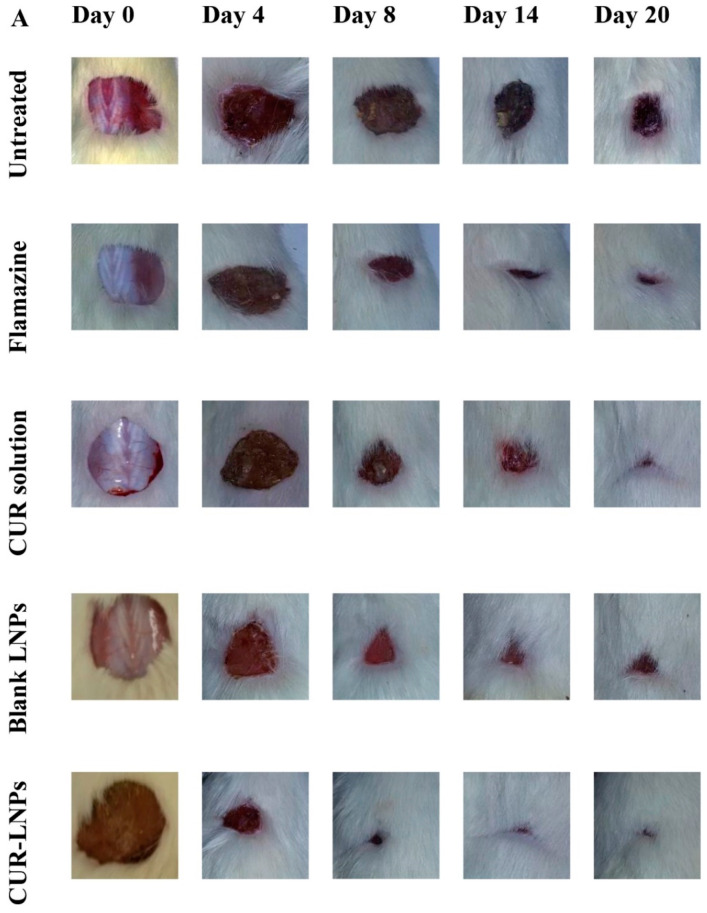
Wound healing efficacy of CUR-loaded LNPs compared to untreated (control), Flamazine, CUR solution, and blank LNPs. Adopted with permission from [132], Elsevier, 2020.

**Table 1 pharmaceutics-15-00657-t001:** A summary of natural polymers-based PNPs used for the treatment of different topical skin diseases.

Type of Skin Disease	Type of Polymer(s)	Active Ingredient(s)	Drug Delivery System Design	PS, ZP	Study Design	Major Findings	Ref.
Psoriasis	CS	TAC	TAC-CS-NPs	PS: 140.8 ± 50.0 nm	In vitro: Skin deposition study. In vivo: Anti-psoriatic efficacy on mice with psoriatic plaques.	Greater drug retention in skin compared to marketed TAC ointment. Skin recovery represented by increased hair growth rate.	[20]
Psoriasis	CS/ALG	CUR	CUR-CS/ALG PNPs combined with photo-irradiation	PS: 200–300 nm ZP: −20 to −30 mV	In vitro (normal and TNF-α-induced cultured HaCaT cells): Cytotoxicity assay. Anti-proliferation studies.	CUR-CS/ALG-NPs showed no toxicity against normal HaCaT cells. A significant increase in anti-proliferative efficacy against TNF-α-induced keratinocytes hyper-proliferation.	[25]
Psoriasis	CS	CUR	CUR-CS-NPs incorporated in collagen-based patch	PS: 164 ± 52 nm ZP: +38 mV	In vitro: Swelling ratio and drug release kinetics. Anti-proliferative activity on psoriatic human keratinocytes obtained from patient donors.	Swelling ratio of the system reached up to 1500% with an initial burst release of CUR followed by a sustained release. Anti-proliferative activity was observed in in vitro psoriatic keratinocytes with maintained cell viability.	[32]
Psoriasis	CS	EGCG	EGCG-CS-NPs	PS: 80–225 nm ZP: 38.3 mV	In vitro: Anti-psoriatic effect on keratinocytes. In vivo: Evaluation of scales and erythema, ear thickness, infiltration of immune cells, and angiogenesis (IMQ-induced skin inflammation in BALB/c mice).	2–4-fold higher inhibition of cell growth in case of PNPs compared to free drug. Reduction in ear and skin thickness, infiltration of immune cells, and decreased proliferation.	[33]
Psoriasis	CS	Gallic acid and rutin	Gallic acid and rutin-loaded CS-based Tween 80-coated PNPs	-	In vitro Proliferation of HaCaT cell (human keratinocytes).	Marked repression of keratinocyte proliferation and significant improvement in anti-inflammatory efficacy.	[34]
Psoriasis	CS	FUCO	FUCO-NLCs coated with CS	PS: 250–400 nm	In vitro: Cell uptake study in normal dermal human fibroblasts.	Significant improvement in FUCO uptake in fibroblasts with greater reduction in viability of psoriatic cells.	[35]
AD	CS	BMV	BMV-CS-NPs	PS: <250 ± 28 nm ZP: +58 ± 8 mV	In vitro: Release kinetics. Drug permeation through full-thickness skin and % drug retained into different skin layers.	Drug displayed a Fickian diffusion type of release (pH 5.5). Higher drug permeation efficiency and greater amount of BMV retained into the skin case of BMV-CS-NPs.	[36]
AD	CS	BMV	HA-BMV-CS-NPs	PS: <300 ± 28 nm ZP: +58 ± 8 mV	In vitro and ex vivo: Drug release studies. Drug penetration through full-thickness skin. Skin deposition study.	Drug displayed a Fickian diffusion type of release (pH 5.5). HA-functionalization significantly improves BMV permeation efficiency through the full-thickness mouse skin as well as amount of BMV retained in the epidermis and the dermis.	[37]
AD	CS	HC and HT	HC-HT-CS-NPs AQ cream	PS: <250 nm	In vivo: Safety testing on healthy human skin to measure TEWL, erythema intensity, and irritancy score. Hematological testing and ACTH levels to evaluate systemic toxicity.	No signs of local irritation, redness, and toxicity were observed. No signs of systemic toxicity.	[38]
AD	CS	HC	HC-CS-NPs	PS: 214 ± 12 nm ZP: +40 ± 4 mV	In vivo (AD-induced NC/Nga mouse model): Anti-inflammatory efficacy. Immunomodulatory efficacy. Histological observation.	Superior anti-inflammatory efficacy in case of HC-CS-NPs in both QV and aqueous-based formulations compared to control and marketed formulation. Reduction in the release of inflammatory mediators. Long and thin elastic fibers observed in animals treated with HC-CS-NPs (skin recovery).	[39]
ACD	CS	HC	HC-loaded CS-NPs (HC-CS-NPs)	PS: 382 ± 14 to 187 ± 12 nm ZP: +14 ± 2 to +45 ± 4 mV	Ex vivo (dermatome mouse skin): Drug permeation studies. Amount of HC retained in skin layers. In vivo: Clinical efficacy in 2,4-DNFB-induced NC/Nga mouse model.	Greater HC permeation in case of NPs-based treatment group compared to control group. Two-fold increase in HC amount retained in the epidermis and dermis. Stronger anti-AD efficacy (lower TEWL, skin thickness, erythema, dryness, exudates, and erosion) compared to commercial formulation.	[40]
AD	CS	HC and HT	HC/HT co-loaded CS-NPs (HC-HT-CS-NPs)	PS: 244 ± 21 nm ZP: +38 ± 4 mV	In vivo (2,4-DNFB-induced AD NC/Nga mouse model: Clinical efficacy. Immunological studies. Histological examination.	Superior anti-AD efficacy observed in animals treated with HC/HT-CS-NPs-based topical cream. Significantly higher repression in the expression of IgE, histamine, PGE2, VEGF-α, Th1, and Th2 in mice skin homogenates treated with the NPs. Faster recovery of skin infrastructure.	[41]
AD	CS	TAC	TAC-CS-HA NPs	PS: 117 ± 19 nm ZP: +63.8 ± 6.4 mV	In vitro: Drug release. Ex vivo: Drug permeation and percentage of drug retained in the epidermis and dermis in NC/Nga mouse skin. In vivo: Therapeutic efficacy assessed by evaluating TEWL, erythema intensity, and ADI in NC/Nga mice.	Biphasic release with relatively sustained release initially followed by rapid release. In case of HA-TAC-CS-NP, lower penetration of TAC, larger amount of drug deposited in the epidermis and dermis, lower TEWL, erythema intensity, and ADI.	[42]
AD	GG	NA	GG-PNPs	PS: 80 nm	In vitro: Wound healing study in NIH3T3 cells. In vivo: Therapeutic effect study on AD-induced Balb/c mice ears, blood cell count, and serum IgE estimation.	Superior anti-AD efficacy. Reduced ear thickness and redness, decreased eosinophils, macrophages and neutrophils, and reduced serum IgE levels.	[29]
Skin cancer	CS and poly(NVPAI) copolymer	5-FU	5-FU-CS-poly(NVPAI) copolymer nanocapsules incorporated in sodium ALG- and HA-based gel	-	Ex vivo: Permeation through chicken skin membrane. In vitro: Hemolysis assay; skin irritation test; and cytotoxicity assay on TE 354.T. In vivo test using male albino mice.	Significant improvement in permeability of 5-FU through the chicken skin membrane. Good compatibility with blood. Significant reduction in cancer cells viability.	[43]
Skin cancer	Chitin	5-FU	FCNGs	PS: 120–140 nmZP: +31.9 mV	In vitro: Drug release. Cytotoxicity against A375 and HDF. Blood compatibility. Ex vivo: Skin permeation and retention (porcine skin). Histological evaluation.	pH-responsive swelling and drug release. In concentrations of 0.4–2.0 mg/mL, FCNG exhibited significant cytotoxic effects on A375 cells line. Very low hemolytic ratio. 4–5 times higher retention into deeper skin layers.	[44]
Fungal skin infection	CS	VRC	VRC-CS-NPs-FFS	PS: 238 nm	Ex vivo: Skin deposition study in mice skin. Antifungal assay against *C. albicans* and *A. flavus.* In vivo: Local irritation study in albino rabbits.	VRC-CS-NPs showed higher deposition in skin layers. 5.9-fold higher inhibition against *C. albicans* and *A. flavus*. No signs of irritation were observed.	[45]
Fungal skin infection	CS/XG	TB	TB-CS/XG NPs gel	PS: 221.3 nm ZP: +19.51 to +26.23	Ex vivo: Permeation and retention studies in albino rat skin. In vivo: Antifungal studies in albino rats infected topically with *C. albicans.*	Increased permeation and higher retention into skin layers when CS concentration increased and XG concentration was reduced. Significant decrease in fungal bioburden in animals treated with TB-CS/XG NPs compared to control.	[30]
Fungal skin infection	EC	ITZ	ITZ-EC NPs gel	PS: 200 nm	Ex vivo: Permeation study (rat epidermis). In vitro: Antifungal activity against *C. albicans* and *A. niger.*	20-fold higher permeability and 7-fold higher skin retention with ITZ-NPs. Better fungal inhibition against *C. albicans* and *A. niger*.	[46]
Parasitic skin infection	CS	AmB	AmB-CS-TPP or AmB-CS-Dex) PNPs	(AmB-CS-TPP) PS: 69 ± 8 nm ZP: 25.5 ± 1 mV (AmB-CS-Dex) PS: 174 ± 8 nm ZP: −11 ± 1 mV	In vitro: Drug release. Activity against *Leishmania major*. Cytotoxicity assay against KB cells and RBCs. Ex vivo: Drug permeation into infected skin lesions (BALB/c mouse skin).	Sustained release of AmB from both types of PNPs with 1.5 times greater release from CS-TPP-NPs. AmB-PNPs showed significantly lower cytotoxicity against RBCs and KB cells (biocompatible). AmB-PNPs exhibited highest antifungal efficacy compared to free AmB and AmBisome^®^.	[47]
Viral skin infection	CS	ACR	Span 80/TPGS modified ACR-LCNCs	PS: 177.50 ± 1.41 nm ZP: −10.70 ± 0.85 mV	Ex vivo: Permeation and retention study in the skin of Wistar rats. In vivo (healthy Wistar rats): CLSM study. Tolerance and safety study. Dermatokinetics study.	Higher permeation of LCNCs (with Span 80/TPGS) compared to LCNCs (without surfactant blend) and ACR suspension. No erythema with normal histological features in drug-free LCNCs 8 and ACR-loaded LCNCs 8 groups. Higher drug deposition in skin with LCNCs 8 compared to other formulations.	[48]
Bacterial skin infection	Gelatin and PVA	GO/Ag	GO/Ag NPs loaded in gelatin/PVA-based hydrogel	-	In vitro: Biocompatibility assays: cytotoxicity using Vero cells (ATCC^®^ CCL-81), hemolysis, and platelet aggregation. Antimicrobial testing against *E. coli* and *S. aureus.*	Good biocompatibility against Vero cells at low and middle concentrations with minimal hemolytic effect and moderate platelet aggregating capacity. GO-Ag NPs nanoconjugates hydrogel showed 100% inhibition of bacterial growth at 20 μg/mL.	[49]
Skin wound	CS	CUR and REV	HA-CUR-REV-CS-NPs	PS: 138 ± 11 nm ZP: +35.4 ± 1.4 mV	In vitro: Release kinetic studies.	A biphasic release pattern was observed with initial fast release of drugs followed by a continuous sustained release. Followed non-Fickian diffusion type of release.	[28]
Burn wound	CS	CUR and QUE	HA-CUR-QUE-CS-NPs	PS: 177 ± 11 nm ZP: +37.0 ± 3.2 mV	In vitro: Drug release study. Cell proliferation (MC3T3-E1 cells). Ex vivo: Drug permeation and retention into full-thickness rat skin. In vivo: Burn wound healing efficacy (Wistar rat) and histological examination.	HA-functionalized CUR-QUE-CS-NPs exhibited a triphasic release pattern and Fickian diffusion. Higher proliferation rate of MC3T3-E1 cells. Higher permeation efficiency and retention into epidermis and dermis. Superior wound healing (faster wound closure rate) compared to control groups.	[50]
Infected skin wound	CS	Cefepime	Cefepime-CS-NPs embedded in HA-PVA-pullulan-based hydrogel membranes	PS: 172 nm ZP: +27.8 mV	In vitro: Drug release and swelling studies. WVTR and oxygen permeability tests. Antibacterial activity analysis. Cytocompatibility evaluation (HT1080 cell line). In vivo (Sprague Dawley rats): Wound healing or closure rate.	Hydrogel membrane showed good swelling capacity and sustained release of drug. WVTR and oxygen permeability values were within the ideal dressing range. Higher zone of inhibition against *S. aureus*, *E. coli*, and *P. aeruginosa*. No cytotoxicity against HT1080 cells. Accelerated wound healing.	[51]
Acne	CS and HA	Clindamycin	Clindamycin-CS-NPs or Clindamycin-HA NPs	PS (CS): 362 ± 19 nm PS (HA): 417 ± 9 nm	In vitro: Drug deposition study in porcine skin.	Enhanced drug accumulation in PSU compared to marketed formulation with 1.5-fold higher targeting in case of HA NPs compared to CS-NPs.	[27]
Acne	DLX	IST	IST-DLX NPs	PS: 230 nm ZP: negative	In vitro: Skin distribution study in pig ear skin. Inflammatory modulation in AMJ-2 macrophage cells. In vivo: Skin irritation studies in Wistar rats.	3-fold higher accumulation of IST in hair follicles compared to IST solution. Restrained IL-6 expression. Reduced photo-irritation.	[52]
Acne	TPGS	ADA	ADA-TPGS PNPs	PS: <20 nm	In vitro: Finite dose experiments in porcine and human skin. Drug delivery studies in human skin on intact PSU.	Permeation efficiency was 2- and 10-fold greater than marketed ADA gel and cream, respectively. 4.5- and 3.3-fold higher ADA accumulation in PSU compared to marketed ADA gel and cream, respectively.	[53]
Acne	CS/ALG	BP	BP-CS/ALG NPs	PS: 341.6 ± 11.1 nm	In vitro: Antimicrobial activity against *P. acnes.* Anti-inflammatory activity in human monocytes and keratinocytes.	Dose-dependent antibacterial effect against *P. acnes.* Anti-inflammatory effect caused by inhibition of cytokines induced by *P. acnes*.	[54]
Acne	PLGA	TH	TH-PLGA NPs	PS: 162–235 nm ZP: −22 to −31 mV	In vitro and ex vivo: Antimicrobial efficacy on *S. epidermidis* and *C. acnes.* In vitro: Antioxidant efficacy and cytotoxicity in HaCaT cells. Ex vivo: Permeation study in healthy and damaged abdominal human skin.	Antimicrobial activity against *C. acnes* with insignificant effect on *S. epidermidis* preserving the normal microbiota. Improved antioxidant activity in keratinocytes with no cytotoxicity at concentrations up to 50 µg/mL. Higher TH permeation in damaged skin compared to normal skin. Higher amount retained in the skin in TH-NP versus free TH.	[55]
Acne	Pluronic^®^ F127	BP	BP-Pluronic^®^ F127 polymeric micelles	PS: 25.3 ± 0.3 nm ZP: −2.5 mV	In vitro: Skin penetration studies in porcine skin. Cytotoxicity studies on mouse embryonic fibroblast cells.	3-fold higher deposition of BP compared to commercial gel. IC50 was 7.6 higher than SDS.	[56]

Abbreviations: PS: particle size, ZP: zeta potential, CS: chitosan, CUR: curcumin, EGCG: epigallocatechin-3-gallate, TAC: tacrolimus, FUCO: fucoxanthin, NLC: nanostructured lipid carrier, ALG: alginate, PNP: polymeric nanoparticle, HaCaT: human keratinocyte cells, AD: atopic dermatitis, BMV: betamethasone valerate, TNF-α: tumor necrosis factor-α, HA: hyaluronic acid, HC: hydrocortisone, HT: hydroxytyrosol, ACD: allergic contact dermatitis, AQ: aqueous, TEWL: transepidermal water loss, 2,4-DNFB: 2,4-dinitrofluorobenzene, ADI: atopic dermatitis index, ACTH: adrenocorticotropic hormone, IgE: immunoglobulin-E, PGE2: prostaglandin E2, VEGF-α: vascular endothelial growth factor-α, Th1: T helper cell type 1, Th2: T helper cell type 2, GG: guar gum, NA: not applicable, NVPAI: N -vinylpyrrolidone-alt-itaconic anhydride, 5-FU: 5-fluorouracil, FCNG: 5-FU-loaded chitin nanogel, TE 354.T: human basal carcinoma cell line, A375: human melanoma cell line, HDF: human dermal fibroblast cells, VRC: voriconazole, FFS: film-forming spray, XG: xanthan gum, TB: terbinafine, EC: ethylcellulose, ITZ: itraconazole, AmB: amphotericin B, TPP: tripolyphosphate sodium, Dex: dextran sulphate, KB: human squamous carcinoma, RBC: red blood cell, ACR: acyclovir, LCNC: lipid-coated chitosan nanocomplex, CLSM: confocal laser scanning microscopy, GO: graphene oxide, Ag: silver, REV: resveratrol, QUE: quercetin, MC3T3-E1: clonal murine cell line of immature osteoblasts, PVA: polyvinyl alcohol, WVTR: water vapor transmission rate, HT1080: human fibrosarcoma cell line, PSU: pilosebaceous unit, DLX: Delonix, IST: isotretinoin, IL-6: interleukin 6, TPGS: D-α-tocopheryl polyethylene glycol succinate, ADA: adapalene, BP: benzoyl peroxide, PLGA: poly(lactic-co-glycolic acid), TH: thymol, IC50: half maximal inhibitory concentration, SDS: sodium dodecyl sulphate.

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
