# Peer review of "Polymeric Nanoparticles as Tunable Nanocarriers for Targeted Delivery of Drugs to Skin Tissues for Treatment of Topical Skin Diseases"

_pharmaceutics, 2023, doi:10.3390/pharmaceutics15020657_

Round 1

Reviewer 1 Report

The manuscript pharmaceutics-2198975 reviews the current knowledge related to the delivery systems based on the polymeric nanoparticles (PNPs) prepared from natural polymers, which are used for the treatment of different skin diseases, such as psoriasis, atopic dermatitis, skin infections, skin cancer, acute-to-chronic wounds and acne.

In my opinion, the manuscript is logically organized, which makes it easy to read and understand and contains relevant information. However, some improvements must be done before to be published in the Pharmaceutics journal, these being mentioned below:

Title

-There is no such a thing as “Natural Polymeric Nanoparticles”! Nanoparticles are obtained from natural polymers; these cannot be found in nature! Please revise this denotation!

Abstract

-L. 39: In my opinion it is more correct to write “polymeric nanoparticles (PNPs) that are prepared from natural polymers” than “natural polymeric nanoparticles (PNPs)”! Please revise it!

1.1.  Skin structure

- L. 94-97: Figure 1 consists of two figures, which must be explained independently and marked with 1a and 1b! Please revise Figure 1 and its caption! All the explanations from section 1.1 must be correlated with Figure 1!

1.2. Topical drug delivery through the skin

- L. 115: Is a repetition of “for”! Please revise it!

- L. 118-123: Please make the equation visible, put it between two paragraphs, write it with Microsoft Equation Editor and mark it with (1), due to the fact that the equations must be noted consecutively within the manuscript! In this form only the legend is visible!

- L. 118: The reference [10] must be in front of “:”! In the present form is like the authors divide the equation by 10! Please revise it!

- L. 110-123: Please improve this paragraph, to better highlight the differences between topical, dermal and transdermal drug delivery! In addition, please present the advantages and disadvantages of these three actions, presenting the therapeutic effect of the drugs for each type of release!

- Please emphasize the novelty of this manuscript compared to the other articles from literature!

2. Physicochemical features and types of PNPs

- L. 163-164: “or synthetic,” not “or synthetic polymers”!

- L. 164-165: It is not necessary to write “Poly” with majuscules! Please revise both synthetic polymers!

- L. 187-188: Regarding Figure 2, the “adsorbed drug” must be represented in both cases, because it can be adsorbed on the surface of both, the Nanocapsules and the Nanospheres! Please improve Figure 2 with this explanation or delete the "adsorbed drug" from the Nanocapsules, so as not to mislead the readers!

- There is no information about the natural polymers used in the preparation of nanoparticles, or about the methods of obtaining the nanoparticles. Please add a separate section, where to present the natural polymers used in the obtaining of nanoparticles that are discussed in this review, adding the most important characteristics each type of natural polymer. In this way, the authors respond to the title proposed by them for this paper, related to the presentation of "Natural Polymeric Nanoparticles as Tunable Nanocarriers for Targeted Delivery of Drugs to Skin Tissues for Treatment of Topical Skin Diseases"!

- In addition, please briefly present the preparation methods of these nanoparticles, including those from the examples which are further mentioned in section 3 from the manuscript!

3.6.3. PNPs-based topical therapies for acne

- Please add the caption for Table 1!

- In Table 1 the information is presented mixed, either with the full name, or with an abbreviation, or both forms, examples in this sense being: Chitosan (CS) [32], Chitosan [100], Chitosan (CH) [102] and CS [103]! Please use a uniform form of presentation of compounds! If the authors decide to use only abbreviations, please explain them all at the bottom of the table. If not, use the full compound name for each example, without abbreviations! Please revise the entire Table 1!

- Table 1 cannot be added to a sub-section related to acne therapies, "3.6.3. PNPs-based topical therapies for acne", given the fact that it shows nanoparticles used for all diseases! This Table 1 should be moved to a new section, Section 4, where general information on the natural polymers used in the preparation of nanoparticles can be presented, along with the importance of each type of natural polymer or type of nanoparticle for a particular disease, or other conclusions of this study!

Another option would be to move it to a section 3.1, with a possible title: "Natural polymers in PNPs used for treatment of skin diseases", where the natural polymers used in the preparation of PNPs should be described. Please improve the part related to Table 2!

References

-Please revise all the references and write them according to the journal’s requirements, which ca be found in the Instructions for Authors!

Author Response

05 Feb 2023

Prof. Nebojsa Salaj

Associate Editor

Prof. Sabrina Belbekhouche

Guest Editor

Special issue: Major Contribution of Natural Polymers for Biological Applications in the Last

                 10 Years: Toward Tailor-Made Biotechnologies

Pharmaceutics MDPI

Subject: Response to Reviewer’s and Editor’s comments and submission of Revised Manuscript (pharmaceutics-2198975)

Dear Prof/Dr.,

We are pleased to inform you that we have carefully considered all comments from the Referee-1 and the editorial team and have revised our manuscript accordingly. All the changes made in the revised version of the manuscript have been highlighted via track changes (Simple Markup). Additionally, we have performed a through proofreading to the revised manuscript to remove minor spelling, grammar, or punctuation errors.

Following is the list of all comments from the reviewers and our response to their queries and recommendations.

Referee-1

Author’s response

The manuscript pharmaceutics-2198975 reviews the current knowledge related to the delivery systems based on the polymeric nanoparticles (PNPs) prepared from natural polymers, which are used for the treatment of different skin diseases, such as psoriasis, atopic dermatitis, skin infections, skin cancer, acute-to-chronic wounds and acne.

In my opinion, the manuscript is logically organized, which makes it easy to read and understand and contains relevant information. However, some improvements must be done before to be published in the Pharmaceutics journal, these being mentioned below.

We greatly acknowledge Referee 1 for recommending our manuscript for publication. We have carefully considered all comments from referee-1 and have revised our manuscript accordingly.

1.

Title

There is no such a thing as “Natural Polymeric Nanoparticles”! Nanoparticles are obtained from natural polymers; these cannot be found in nature! Please revise this denotation!

Based on refere-1 suggestion, we have revised the title of our manuscript as follow.

Polymeric Nanoparticles as Tunable Nanocarriers for Targeted Delivery of Drugs to Skin Tissues for Treatment of Topical Skin Diseases” (page 1).

2.

Abstract

·       L.39: In my opinion it is more correct to write “polymeric nanoparticles (PNPs) that are prepared from natural polymers” than “natural polymeric nanoparticles (PNPs)”! Please revise it!

We have revised the mentioned sentence as suggested (page 3, line 44-45).

3.

1.1.  Skin structure

·       L. 94-97: Figure 1 consists of two figures, which must be explained independently and marked with 1a and 1b! Please revise Figure 1 and its caption! All the explanations from section 1.1 must be correlated with Figure 1!

Figure 1 has been revised as Figure 1A and 1B. Its figure caption has also been revised. Section 1.1 has also been explained in correlation with Figure 1A & 1B (page 5-7, lines 77-106).

4.

1.2. Topical drug delivery through the skin

·       L. 115: Is a repetition of “for”! Please revise it!

·       L. 118-123: Please make the equation visible, put it between two paragraphs, write it with Microsoft Equation Editor and mark it with (1), due to the fact that the equations must be noted consecutively within the manuscript! In this form only the legend is visible!

·       L. 118: The reference [10] must be in front of “:”! In the present form is like the authors divide the equation by 10! Please revise it!

·       L. 110-123: Please improve this paragraph, to better highlight the differences between topical, dermal and transdermal drug delivery! In addition, please present the advantages and disadvantages of these three actions, presenting the therapeutic effect of the drugs for each type of release!

·       Please emphasize the novelty of this manuscript compared to the other articles from literature!

·       We have revised it and deleted the repeated word (page 7, line 116).

·       We have used Microsoft Equation Editor as suggested to write the equation, placed it between two paragraphs, and marked it with (1) (page 8, line 134).

·       Upon referee-1 suggestion, we have modified the location of reference [10] to be in front of “:” (page 8, line 132)

·       We have made the suggested revisions in the mentioned paragraph (pages 7-8, lines 123-133).

·       Novelty of this manuscript has been explained at the end of section 1.2 (page 9, lines 150-158).

5.

2. Physicochemical features and types of PNPs

·       L. 163-164: “or synthetic,” not “or synthetic polymers”!

·       L. 164-165: It is not necessary to write “Poly” with majuscules! Please revise both synthetic polymers!

·       L. 187-188: Regarding Figure 2, the “adsorbed drug” must be represented in both cases, because it can be adsorbed on the surface of both, the Nanocapsules and the Nanospheres! Please improve Figure 2 with this explanation or delete the "adsorbed drug" from the Nanocapsules, so as not to mislead the readers!

·       There is no information about the natural polymers used in the preparation of nanoparticles, or about the methods of obtaining the nanoparticles. Please add a separate section, where to present the natural polymers used in the obtaining of nanoparticles that are discussed in this review, adding the most important characteristics each type of natural polymer. In this way, the authors respond to the title proposed by them for this paper, related to the presentation of "Natural Polymeric Nanoparticles as Tunable Nanocarriers for Targeted Delivery of Drugs to Skin Tissues for Treatment of Topical Skin Diseases"!

·       In addition, please briefly present the preparation methods of these nanoparticles, including those from the examples which are further mentioned in section 3 from the manuscript!

·       Suggested revision has been made in the revised manuscript (page 10, lines 173-174).

·       Suggested revision (capitalized “Poly”) has been made in the revised manuscript (page 10, lines 173-174).

·       Figure 2 has been modified as suggested. Adsorbed drugs have now shown on both nanospheres as well as nanocapsules (page 11).

·       A separate section 3.1 has been added in the revised manuscript (pages 11-13, lines 197-233) in which we have summarized specific information about natural polymers that have been employed to fabricate PNPs for treatment of skin diseases. Additionally, we have provided brief description of other natural polymers throughout the manuscript.

·       We have included a brief description about methods of preparation of PNPs in section 3.1 (pages 11-13, lines 197-233.

6.

3.6.3. PNPs-based topical therapies for acne

·       Please add the caption for Table 1!

·       In Table 1 the information is presented mixed, either with the full name, or with an abbreviation, or both forms, examples in this sense being: Chitosan (CS) [32], Chitosan [100], Chitosan (CH) [102] and CS [103]! Please use a uniform form of presentation of compounds! If the authors decide to use only abbreviations, please explain them all at the bottom of the table. If not, use the full compound name for each example, without abbreviations! Please revise the entire Table 1!

·       Table 1 cannot be added to a sub-section related to acne therapies, "3.6.3. PNPs-based topical therapies for acne", given the fact that it shows nanoparticles used for all diseases! This Table 1 should be moved to a new section, Section 4, where general information on the natural polymers used in the preparation of nanoparticles can be presented, along with the importance of each type of natural polymer or type of nanoparticle for a particular disease, or other conclusions of this study!

·       Another option would be to move it to a section 3.1, with a possible title: "Natural polymers in PNPs used for treatment of skin diseases", where the natural polymers used in the preparation of PNPs should be described. Please improve the part related to Table 1!

·        Caption for Table 1 has been added as follow (page 48).

“A summary of natural polymers used to prepare PNPs for treated of different skin diseases.”

·       We have revised Table 1 and have only presented abbreviations of drugs and polymers within the Table. The full forms of all abbreviations have been provided at the bottom of the table (page 61-62).

·       Table 1 was not added to section 3.6.3. It is presented at the end of all sections where we explained the efficacy of other types of PNPs used for treatment of skin diseases.

We anticipate that current location of Table 1 is more logical because it contains a lot of references which must be presented in the text of the manuscript before summarizing in the Table 1. Therefore, we suggest keeping it at the current location.

7.

References

·       Please revise all the references and write them according to the journal’s requirements, which can be found in the Instructions for Authors!

We have now formatted all references according to the Instructions for Authors.

Our research group greatly acknowledges Referee-1 insightful remarks and improvement of our manuscript. We anticipate that our response to reviewer’s comments will satisfy their queries and we do hope for your kind consideration to our “Revised Manuscript” for publication in your esteemed journal.

Best regards,

DR. ZAHID HUSSAIN, on behalf of all co-authors

Drug Delivery Research Group

Department of Pharmaceutics & Pharmaceutical Technology, College of Pharmacy, University of Sharjah, Sharjah 27272, United Arab Emirates

E-mail: zhussain@sharjah.ac.ae; zahidh85@yahoo.com

ORCID: https://orcid.org/0000-0002-6978-3133

Reviewer 2 Report

Very comprehensive study regarding the application of polymeric nano-particles in skin diseases management compared to other reviews found in the literature.

Although, I noticed that the vast majority (approx. 75%) of the studies presented are based on chitosan (CS) based nano-particles, alone or in combination with other natural polymers, however the study is comprehensive, describing for each skin disease (psoriasis, atopic dermatitis, skin infections, skin cancer, acne) the pathogenesis, the conventional treatments with their side effects and limitations, as well as the advantages of using systems based on polymeric nano-particles NPs.

The manuscript is very well structured, easy to follow and the data are well summarized in the Table. I agree with its publication after making some changes, namely:

- I think that the Abstract is too long; it contains more than 300 words, so it should be reduced to about 200 words by concentrating the information.

- I recommend paying attention to the abbreviations that appear in text for the first time (e.g. SC, TPGS, ...)

- Line 56: “1. Introduction”: before going directly to “1.1. Skin structure”, at least an preliminary idea should be written about the necessity of the study and what the study will contain. The transition is too sudden.

- Is Figure 1 original?? If it is not original, it should be mentioned where it was taken from.

- Line 118: The equation that describes the Fick's second law must be made more visible, it is very crowded in the text, it is barely visible.

- Line 168: The range specified for nanometer sizes is too wide “1-1000nm”. A range between 0 and 100 nm is usually accepted for nano-scaled dimensions.

- Line 271: ” Under the neutral and acidic conditions, the amino groups of CS undergo protonation…” - the amino groups of chitosan become protonated only in acidic pH, not neutral, otherwise chitosan would be soluble in neutral medium, which is not true. So I recommend deleting "neutral" from the sentence.

- Line 292: IMQ - ?

- Line 308: In Figure 4M,N,O,P,Q: what does it mean *, **, *** ?

- Line 347: Pay attention to the name of the nanoparticles, it is CS/ALG-NPs and not ALG/CS-NPs.

- Line 348: “These findings…” it is repeated also in the previous sentence, it should be replaced with a synonym.

- Again, attention to the abbreviations that appear for the first time in the text: Line 361: IgE - ? ; Line 431: DNFB - ?; Line 529: PpIX - ? ; Line 680: RBCs - and KB - ?;

- Line 830: “tensile strength, porosity, mechanical properties, and sustained release behavior.” – The tensile strength is included in the mechanical properties, so I don't see why it should be mentioned separately.

Table 1:

- I recommend using the abbreviation for the name of the polymer or active molecule, in order to make the table content easier to read. In this way, columns 1, 2 and 3 are reduced and there is more space for the other columns with larger content.

- In the "Major findings" column, I recommend summarizing the information. There are almost the same sentences as in the main text and it is quite difficult to read.

- Also, the abbreviations can be explained at the end of the table.

- Pay attention to abbreviations, I recommend keeping the same abbreviations: CUR or Cur ?; CS or CH for chitosan?

The References are quite current. However, I noticed some references before the year 2000. I recommend replacing them with studies that are as current as possible. Examples: reference [110] from 1995; [132] from 1989; [135] - 1973; [136] - 1982). I think the authors can find similar studies carried out after the 2000s or even much newer.

Also, I recommend arranging the References according to the Instructions for authors.

Author Response

05 Feb 2023

Prof. Nebojsa Salaj

Associate Editor

Prof. Sabrina Belbekhouche

Guest Editor

Special issue: Major Contribution of Natural Polymers for Biological Applications in the Last

                 10 Years: Toward Tailor-Made Biotechnologies

Pharmaceutics MDPI

Subject: Response to Reviewer’s and Editor’s comments and submission of Revised Manuscript (pharmaceutics-2198975)

Dear Prof/Dr.,

We are pleased to inform you that we have carefully considered all comments from the Referee-2 and have revised our manuscript accordingly. All the changes made in the revised version of the manuscript have been highlighted via track changes (Simple Markup). Additionally, we have performed a through proofreading to the revised manuscript to remove minor spelling, grammar, or punctuation errors.

Following is the list of all comments from the reviewers and our response to their queries and recommendations.

Referee-2

Author’s response

Very comprehensive study regarding the application of polymeric nanoparticles in skin diseases management compared to other reviews found in the literature.

Although, I noticed that the vast majority (approx. 75%) of the studies presented are based on chitosan (CS) based nano-particles, alone or in combination with other natural polymers, however the study is comprehensive, describing for each skin disease (psoriasis, atopic dermatitis, skin infections, skin cancer, acne) the pathogenesis, the conventional treatments with their side effects and limitations, as well as the advantages of using systems based on polymeric nano-particles NPs.

The manuscript is very well structured, easy to follow and the data are well summarized in the Table. I agree with its publication after making some changes, namely:

We greatly appreciate Referee-2 for recommending our manuscript for publication. We have carefully considered all the comments from referee-2 and have revised our manuscript accordingly.

1.

I think that the Abstract is too long; it contains more than 300 words, so it should be reduced to about 200 words by concentrating the information.

Upon reviewer suggestion, we have now revised the Abstract and reduced it to approximately 212 words (page 3, lines 38-54).

2.

I recommend paying attention to the abbreviations that appear in text for the first time (e.g. SC, TPGS, ...)

We have thoroughly revised our manuscript and ensured that each abbreviation appears in the text with full form at its first instances followed by using the abbreviation thereafter.

3.

Line 56: “1. Introduction”: before going directly to “1.1. Skin structure”, at least a preliminary idea should be written about the necessity of the study and what the study will contain. The transition is too sudden.

We have now added a small introductory paragraph for a smooth transition to the sub-section 1.1 of the introduction (page 5, lines 65-74).

4.

Is Figure 1 original?? If it is not original, it should be mentioned where it was taken from.

Yes Figure 1 is original and is created with BioRender.com and it is mentioned in the figure legend.

5.

Line 118: The equation that describes the Fick's second law must be made more visible, it is very crowded in the text, it is barely visible.

We have used Microsoft Equation Editor as suggested to write the equation, placed it between two paragraphs, and marked it with (1) (page 8, line 134).

6.

Line 168: The range specified for nanometer sizes is too wide “1-1000nm”. A range between 0 and 100 nm is usually accepted for nano-scaled dimensions.

In the mentioned sentence, we aimed to define nanomaterials (materials having dimensions 1–1000 nm) regardless of the influence of size on the efficacy of PNPs. We understand that relatively smaller PNPs exhibit better permeation and efficacy compared to larger particles. Hence, to avoid reader’s confusion, we suggest keeping the original definition of nanoparticles.

7.

Line 271: ” Under the neutral and acidic conditions, the amino groups of CS undergo protonation…” - the amino groups of chitosan become protonated only in acidic pH, not neutral, otherwise chitosan would be soluble in neutral medium, which is not true. So I recommend deleting "neutral" from the sentence.

We have now revised the sentence and deleted the word “neutral” from it (page 16, line 302).

8.

Line 292: IMQ - ?

The full form of IMQ (imiquimod) has now written in the revised manuscript (page 17, line 316).

9.

Line 308: In Figure 4M,N,O,P,Q: what does it mean *, **, *** ?

For better understanding of the readers, we have now mentioned the meanings of *,**,***, and **** in the Figure 4 legend (page 18, lines 333-336).

10.

Line 347: Pay attention to the name of the nanoparticles, it is CS/ALG-NPs and not ALG/CS-NPs.

We have corrected the typo error in the name of the nanoparticles (page 19 lines 353, 355).

11.

Line 348: “These findings…” it is repeated also in the previous sentence, it should be replaced with a synonym.

We have revised the sentence (page 22, line 407).

12.

Again, attention to the abbreviations that appear for the first time in the text: Line 361: IgE - ? ; Line 431: DNFB - ?; Line 529: PpIX - ? ; Line 680: RBCs - and KB - ?;

We have thoroughly revised our manuscript and ensured that all abbreviations appear in the text with full forms at their first instances followed by using the abbreviations throughout the manuscript.

13.

Line 830: “tensile strength, porosity, mechanical properties, and sustained release behavior.” – The tensile strength is included in the mechanical properties, so I don't see why it should be mentioned separately.

We have revised the sentence as follow (page 39, line 763-765).

“In vitro characterization showed that developed hydrogel membrane exhibited good swelling characteristics, porosity, mechanical properties, and sustained release behavior”

14.

Table 1

I recommend using the abbreviation for the name of the polymer or active molecule, in order to make the table content easier to read. In this way, columns 1, 2 and 3 are reduced and there is more space for the other columns with larger content.

We have revised the Table 1 and have only presented abbreviations of drugs and polymers within the Table. The full forms of all abbreviations have been provided at the bottom of the table (pages 61-62).

15.

In the "Major findings" column, I recommend summarizing the information. There are almost the same sentences as in the main text and it is quite difficult to read.

We have summarized the column “Major findings” in the Table 1.

16.

Also, the abbreviations can be explained at the end of the table.

The full forms of all abbreviations have been provided at the bottom of the table (pages 61-62).

17.

Pay attention to abbreviations, I recommend keeping the same abbreviations: CUR or Cur ?; CS or CH for chitosan?

We have thoroughly reviewed our manuscript and ensured that all abbreviations and uniform (e.g., CUR for curcumin, and CS for chitosan).

18.

The References are quite current. However, I noticed some references before the year 2000. I recommend replacing them with studies that are as current as possible. Examples: reference [110] from 1995; [132] from 1989; [135] - 1973; [136] - 1982). I think the authors can find similar studies carried out after the 2000s or even much newer.

We have replaced 1900s references with more recent ones, now all references are after the year 2000.

19.

Also, I recommend arranging the References according to the Instructions for authors.

We have now formatted all references according to the Instructions for Authors.

Our research group greatly acknowledges Referee 2 insightful remarks and improvement of our manuscript. We anticipate that our response to reviewer’s comments will satisfy their queries and we do hope for your kind consideration to our “Revised Manuscript” for publication in your esteemed journal.

Best regards,

DR. ZAHID HUSSAIN, on behalf of all co-authors

Drug Delivery Research Group

Department of Pharmaceutics & Pharmaceutical Technology, College of Pharmacy, University of Sharjah, Sharjah 27272, United Arab Emirates

E-mail: zhussain@sharjah.ac.ae; zahidh85@yahoo.com

ORCID: https://orcid.org/0000-0002-6978-3133

Reviewer 3 Report

The review manuscript from Hussain and coworkers mainly focused on biopharmaceutical viability of natural PNPs-based drug delivery systems for topical administration and treatment of various skin diseases such as psoriasis, atopic dermatitis, skin infections, skin cancer, acute-to-chronic wounds, and acne. A good review should at least contain: (1) summarization of the published work, (2) outline the key points for the topic, (3) point out the advances and drawback, as well as (4) reasonable outlook. As a matter of fact, a large number of PNPs have been developed for the treatment of skin diseases including many intelligent systems. This requires the authors to highlight the uniqueness of these PNPs in clinical applications and possible treatment mechanisms, in the writing of this manuscript rather than spending a lot of words in their therapeutic effects. This will be helpful for subsequent researchers to develop PNPs with better performance. Nevertheless, this work should be interested to the readers of Pharmaceutics, but I recommend publication after major revision described below. Specifically, I have asked for clarification of details and interpretation of PNPs.

1.      As the described “Depending on their morphology and architecture, PNPs can be classified as nanocapsules or nanospheres (Figure 2)”. However, such as the polymeric nanogels have been found as the attractive drug carrier system and used for the treatment of skin diseases. Therefore, the features and types of PNPs were descripted not comprehensively. I suggest author re-write all corresponding part.

2.      As the title of this review “Natural Polymeric Nanoparticles as Tunable Nanocarriers for Targeted Delivery of Drugs to Skin Tissues for Treatment of Topical Skin Diseases”. However, the “natural polymeric hydrogel membrane” are mentioned several times in following parts in this manuscript. The similarities and differences between the polymeric nanoparticles and polymeric hydrogel membrane must be explained clearly.

3.      As the described “Drug release happens after the degradation of the matrix, thus controlling the polymer’s degradation can help in controlling the drug release.” This is not proper, it doesn't mean that the drug release occur just after degradation of matrix, since the drug release also occur during the degradation of matrix. In addition, there many ways for the controlling of drug release, such as pH, temperature, etc., which makes the degradation unnecessary. These must be clarified in all corresponding parts to avoid misunderstanding.

4.      The format and writing of this manuscript should be carefully polished and improved to correct, especially to the English writing/expressions, and some errors should be corrected, such as the abbreviations of curcumin (Cur) in line 279 to 347 are different from the one in line 771.

Author Response

05 Feb 2023

Prof. Nebojsa Salaj

Associate Editor

Prof. Sabrina Belbekhouche

Guest Editor

Special issue: Major Contribution of Natural Polymers for Biological Applications in the Last

                 10 Years: Toward Tailor-Made Biotechnologies

Pharmaceutics MDPI

Subject: Response to Reviewer’s and Editor’s comments and submission of Revised Manuscript (pharmaceutics-2198975)

Dear Prof/Dr.,

We are pleased to inform you that we have carefully considered all comments from the Referee-3 and have revised our manuscript accordingly. All the changes made in the revised version of the manuscript have been highlighted via track changes (Simple Markup). Additionally, we have performed a through proofreading to the revised manuscript to remove minor spelling, grammar, or punctuation errors.

Following is the list of all comments from the reviewers and our response to their queries and recommendations.

Referee-3

Author’s response

The review manuscript from Hussain and coworkers mainly focused on biopharmaceutical viability of natural PNPs-based drug delivery systems for topical administration and treatment of various skin diseases such as psoriasis, atopic dermatitis, skin infections, skin cancer, acute-to-chronic wounds, and acne. A good review should at least contain: (1) summarization of the published work, (2) outline the key points for the topic, (3) point out the advances and drawback, as well as (4) reasonable outlook. As a matter of fact, a large number of PNPs have been developed for the treatment of skin diseases including many intelligent systems. This requires the authors to highlight the uniqueness of these PNPs in clinical applications and possible treatment mechanisms, in the writing of this manuscript rather than spending a lot of words in their therapeutic effects. This will be helpful for subsequent researchers to develop PNPs with better performance. Nevertheless, this work should be interested to the readers of Pharmaceutics, but I recommend publication after major revision described below. Specifically, I have asked for clarification of details and interpretation of PNPs.

We greatly appreciate Referee-3 feedback and suggested revisions to improve our manuscript. We have carefully considered all the comments of referee-3 and have revised our manuscript accordingly.

1.

As the described “Depending on their morphology and architecture, PNPs can be classified as nanocapsules or nanospheres (Figure 2)”. However, such as the polymeric nanogels have been found as the attractive drug carrier system and used for the treatment of skin diseases. Therefore, the features and types of PNPs were descripted not comprehensively. I suggest author re-write all corresponding part.

Upon referee-3 query, we would like to clarify that polymeric NPs can only be classified into two types, 1) nanospheres and 2) nanocapsules based on the morphology. However, the study referred by the reviewer 3 is about hybrid nanogel (gel containing PNPs), not the conventional nanogel. In this study, researchers fabricated PNPs and then embedded them into polymeric gel for better permeation through the skin and prolonged residential time on the skin.

2.

As the title of this review “Natural Polymeric Nanoparticles as Tunable Nanocarriers for Targeted Delivery of Drugs to Skin Tissues for Treatment of Topical Skin Diseases”. However, the “natural polymeric hydrogel membrane” are mentioned several times in following parts in this manuscript. The similarities and differences between the polymeric nanoparticles and polymeric hydrogel membrane must be explained clearly.

Upon reviewer-3 query, we would like to highlight polymeric hydrogel membrane was used as a vehicle in which drug-loaded PNPs were embedded, and this type of delivery system is called hybrid delivery system. The reason for embedding of PNPs within hydrogel membrane was to improve the residential time of PNPs at the wounded skin, to enhance permeation flux, and improving the wound healing efficacy.

To avoid reader’s confusion, we have now provided sufficient details about this study in the revised manuscript (page 38, lines 733-758).

3.

As the described “Drug release happens after the degradation of the matrix, thus controlling the polymer’s degradation can help in controlling the drug release.” This is not proper, it doesn't mean that the drug release occur just after degradation of matrix, since the drug release also occur during the degradation of matrix. In addition, there many ways for the controlling of drug release, such as pH, temperature, etc., which makes the degradation unnecessary. These must be clarified in all corresponding parts to avoid misunderstanding.

We completely agree with reviewer-3 that release kinetics from the polymeric nanoparticles is very much dependent on the nature, type, and characteristic of the polymer used to fabricate them. Therefore, we have revised the mentioned sentence as follow.

Depending upon the natural and characteristics of the polymer used for preparation of PNPs, drug is released through the diffusion process either due to desorption of adsorbed drug, swelling of the polymeric matrix in response to pH, temperature, light, magnetic field, etc. or degradation of the polymeric matrix [15,17]. Generally, drugs adsorbed on the surface of PNPs results in initial burst release followed by a relatively slow and sustained release of drug from the polymeric matrix or the core (pages 10-11, lines 185-192).

4.

The format and writing of this manuscript should be carefully polished and improved to correct, especially to the English writing/expressions, and some errors should be corrected, such as the abbreviations of curcumin (Cur) in line 279 to 347 are different from the one in line 771.

We have performed a through proofread to our revised manuscript to remove minor spelling, grammar, or punctuation errors. Moreover, we ensure that all abbreviations have been written in full forms on their first appearance in the text.

Our research group greatly acknowledges Referee 3 insightful remarks and improvement of our manuscript. We anticipate that our response to reviewer’s comments will satisfy their queries and we do hope for your kind consideration to our “Revised Manuscript” for publication in your esteemed journal.

Best regards,

DR. ZAHID HUSSAIN, on behalf of all co-authors

Drug Delivery Research Group

Department of Pharmaceutics & Pharmaceutical Technology, College of Pharmacy, University of Sharjah, Sharjah 27272, United Arab Emirates

E-mail: zhussain@sharjah.ac.ae; zahidh85@yahoo.com

ORCID: https://orcid.org/0000-0002-6978-3133

Round 2

Reviewer 1 Report

The authors have responded to all comments and thus, the paper is accepted for publication in its present form.

Reviewer 3 Report

Authors have addressed my concerns properly, and I feel the manuscript is suitable for publishing in its present state.